# Conformational cycle of human polyamine transporter ATP13A2

Jianqiang Mu[1,6], Chenyang Xue [1,6], Lei Fu[2,6], Zongjun Yu [1], Minhan Nie[3], Mengqi Wu[1], Xinmeng Chen[1], Kun Liu[1], Ruiqian Bu[1], Ying Huang[1], Baisheng Yang[1], Jianming Han[1], Qianru Jiang[1], Kevin C. Chan [2], Ruhong Zhou [2], Huilin Li[3,4], Ancheng Huang [1], Yong Wang [2,5] ✉ & Zhongmin Liu [1] ✉

Dysregulation of polyamine homeostasis strongly associates with human diseases. ATP13A2, which is mutated in juvenile-onset Parkinson's disease and autosomal recessive spastic paraplegia 78, is a transporter with a critical role in balancing the polyamine concentration between the lysosome and the cytosol. Here, to better understand human ATP13A2-mediated polyamine transport, we use single-particle cryo-electron microscopy to solve high-resolution structures of human ATP13A2 in six intermediate states, including the putative E2 structure for the P5 subfamily of the P-type ATPases. These structures comprise a nearly complete conformational cycle spanning the polyamine transport process and capture multiple substrate binding sites distributed along the transmembrane regions, suggesting a potential polyamine transport pathway. Integration of high-resolution structures, biochemical assays, and molecular dynamics simulations allows us to obtain a better understanding of the structural basis of how hATP13A2 transports polyamines, providing a mechanistic framework for ATP13A2-related diseases.

Ubiquitously distributed polyamines are essential for many biological processes, including cell proliferation, differentiation, neuroprotection, and apoptosis[1,2]. Polyamine homeostasis is essential for cellular viability and cell growth[3,4], and relies on two major systems: the metabolic system, which regulates the generation and degradation of polyamines, and the transport system, which determines cellular polyamine distribution[5]. Accurate regulation of polyamine homeostasis in human cells is necessary to maintain cellular fitness[6].

As an essential compartment in the polyamine transport system, the lysosome plays a critical role in balancing polyamine concentrations between the cytosol and lysosome[5]. The lysosomal transporter ATP13A2 was reported to export polyamines from the lysosome to the cytosol[7]. Clinically, more than 20 mutations of human ATP13A2 (hATP13A2) have been directly associated with the development of hereditary diseases like Kufor–Rakeb Syndrome (KRS; a type of juvenile-onset Parkinson's disease) and autosomal recessive spastic paraplegia 78 (SPG78)[8–11]. Fibroblasts derived from KRS patients or ATP13A2-deficient cells exhibit abnormal lysosomal behaviors, such as polyamine accumulation, decreased lysosomal membrane stability, lysosomal alkalization, and decreased autophagosome degradation ability[7,12,13]. Conversely, overexpression of hATP13A2 alleviated Parkinson's disease-related neuron damage caused by the accumulation of toxic α-synuclein proteins[14,15]. Together, these findings suggest that ATP13A2 plays a critical role in the polyamine transport system.

[1]Department of Immunology and Microbiology, School of Life Sciences, Southern University of Science and Technology, 518055 Shenzhen, Guangdong, China. [2]Shanghai Institute for Advanced Study, Institute of Quantitative Biology, College of Life Sciences, Zhejiang University, 310027 Hangzhou, China. [3]School of Pharmaceutical Sciences, Sun Yat-sen University, No.132 Wai Huan Dong Lu, Guangzhou Higher Education Mega Center, 510006 Guangzhou, China. [4]Guangdong Key Laboratory of Chiral Molecule and Drug Discovery, School of Pharmaceutical Sciences, Sun Yat-sen University, 510006 Guangzhou, Guangdong, China. [5]The Provincial International Science and Technology Cooperation Base on Engineering Biology, International Campus of Zhejiang University, 314400 Haining, China. [6]These authors contributed equally: Jianqiang Mu, Chenyang Xue, Lei Fu. ✉e-mail: yongwang_isb@zju.edu.cn; liuzm@sustech.edu.cn

The P-type ATPases can be classified into five subfamilies (P1 to P5). The P1, P2, and P3 ATPases are ion pumps; the P4 subfamily comprises lipid flippases;[16] and the P5 ATPases, which are found only in eukaryotic species, can be further divided into type A and B subgroups[17]. P5A was identified as a transmembrane helix dislocase[18], whereas P5B member ATP13A2 functions as a polyamine transporter[7,19]. The P-type ATPases transport substrates by coupling ATP hydrolysis with protein phosphorylation states and adopting a series of intermediate states, consisting of E1, E1-ATP, E1P-ADP, E1P, E2P, E2-Pi, and E2, in a process known as the Post-Albers reaction[17].

The acquisition of high-resolution structures of hATP13A2 throughout the Post-Albers cycle will be crucial to understanding the mechanism of polyamine transport. Although the E1 and E2 states are the two main conformations of the enzyme[17], the dynamic equilibrium between them makes it challenging to capture the structure of hATP13A2 in a single ATP/ADP-free state. Recently, several intermediate structures of hATP13A2 and its yeast homolog YPK9 were solved using single-particle cryo-electron microscopy (cryo-EM)[20–24]. However, the E2 state, a crucial intermediate step for polyamine release, has not yet been captured, and the nominal E1-apo structure recently obtained by introducing two amino acid mutations in the polyamine-binding site[22] may not accurately represent the physiological E1 state of ATP13A2. Therefore, the underlying mechanism of hATP13A2-mediated polyamine transport remains elusive.

ATP13A2 promotes the fusion of the lysosome and autophagosome at the cellular level by recruiting HDAC6 to deacetylate the cytoskeleton protein contacin[25]. Phosphatidylinositol 3,5-bisphosphate (PI(3,5)P$_2$) was also reported to promote the fusion of the lysosome and autophagosome in earlier studies[26]. Furthermore, the lipids phosphatidic acid and PI(3,5)P$_2$ can initiate the activity of ATP13A2 through interaction with its N-terminal region[27]. Positively charged polyamines, the only proven ATP13A2 protein substrates, can interact with phospholipids via electrostatic interactions[28]. These findings suggest that lipids may actively regulate ATP13A2 transportation of polyamines via a mechanism that remains mysterious.

Here, we report seven cryo-EM structures of hATP13A2 in six intermediate states, importantly including the putative E2 and E1-like states, thereby providing a nearly complete atomic-level conformational cycle spanning the Post-Albers reaction process. Integration of high-resolution structures, biochemical assays, and molecular dynamics (MD) simulations provide a structural basis for polyamine substrate recruitment, translocation, and release by human ATP13A2, which will facilitate our understanding of the pathogenic mechanism of hATP13A2 in neuron diseases.

## Results

### Intermediate states of hATP13A2

To study the hATP13A2-based polyamine transport mechanism, 2×Strep-tagged hATP13A2 was transiently expressed in and purified from the HEK293F cell line, yielding a high-quality recombinant protein (Supplementary Fig. 1a, b). In an ATPase activity assay, the purified hATP13A2 protein exhibited high ATP hydrolysis activity, which was notably increased by the presence of polyamines (Supplementary Fig. 1c). To explore the structural basis of hATP13A2-mediated polyamine transport, we used single-particle cryo-EM to determine a series of hATP13A2 structures along the Post-Albers process. Movies of the hATP13A2 protein captured in different intermediate states were recorded and processed with cryoSPARC. After motion correction and contrast transfer function (CTF) estimation, cryo-EM maps were generated of different intermediate states at overall resolutions of 3.2–5.6 Å according to the gold-standard Fourier shell correlation 0.143 criteria (Fig. 1a and Supplementary Figs. 2–8). These high-resolution cryo-EM maps allowed us to accurately build atomic models of most regions of hATP13A2 (Supplementary Fig. 3h), except the region from E114 to R164 in the N-terminal domain (NTD) due to its

poor density (Supplementary Fig. 4h). The overall structure of hATP13A2 has a canonical P-type ATPase architecture, with the cytosolic A domain linking to TM1 and TM2 domains, and the N and P domains sitting on the TM4 and TM5. These domains coordinate to hydrolyze ATP, and TM helix movement facilitates translocation of the substrates (Fig. 1b–d). Notably, TM4 is divided into two short helices (TM4a and TM4b) by the conserved PPALP motif (Fig. 1b and Supplementary Fig. 9), which is essential for substrate transportation by P5B-type ATPases[29]. A unique NTD adjacent to the A domain contains three short hydrophobic helices (denoted H1, H2, and H3) in a triangular shape embedded in the cytosolic leaflet of the lipid bilayer (Fig. 1b, c).

Previous studies obtained high-resolution structures of other P-type ATPases in the E1 state by vitrifying the wild-type ATPase directly[18,30–32]. However, we acquired a final EM map of hATP13A2 in the E1-like state at a resolution of approximately 5.6 Å with the same preparation protocol as that in previous studies[21] (Supplementary Fig. 2) by processing more than 10,000 movies with cryoSPARC[33]. Capturing the hATP13A2 structure in the E1-like state at 5.6 Å enabled us to assign the backbone of the amino acid sequence to the EM map. The structural comparison showed that hATP13A2 in the E1-like state presents a conformation similar to the E1-ATP structure (Supplementary Figs. 10 and 11a), with the cytosolic domains next to each other. As a results, our E1-like structure differs from the recently reported nominal E1 structures (Protein Data Bank (PDB): 7N75, 7N76, and 7FJM), which reassemble E1P-like structures with a root mean square deviation (RMSD) of 2.91 Å (Supplementary Figs. 10 and 11b)[21,22].

We next used established protocols to stabilize P-type ATPases in nucleotide-binding or phosphorylation states[34]. Adenine nucleotides (AMP-PCP or ADP) or phosphate analogs (BeF$_3^-$ or AlF$_4^-$) were used to stabilize hATP13A2 in the E1-ATP or E1P/E2P intermediate states. In the map obtained after incubation with AMP-PCP (an ATP analog), a blob of density was clearly visualized near the KGSPE motif in the N-domain (Fig. 1e, f and Supplementary Fig. 3i), suggesting the conformation of hATP13A2 was successfully trapped in the E1-ATP state. However, in the structures obtained after incubation with BeF$_3^-$ and AlF$_4^-$, no significant difference was observed. By comparing with known E2P structures, we suggest that both BeF$_3^-$ and AlF$_4^-$ incubated structures are in the same E2P state (Supplementary Figs. 10 and 11c).

### Polyamine entry

Recruitment of polyamines to hATP13A2 is the first step of substrate transport from the luminal side to the cytosolic side (Fig. 1a). Accordingly, we found a polyamine entrance binding pocket positioned in the endo-/lysosomal luminal leaflet with an outward-opening architecture in the E2P structure of hATP13A2 (Fig. 2a–c). Spatially, the entrance site (denoted as Site$_1$) located among TM1, TM2, TM4a, TM5, and TM6, had a chalk-shaped architecture open to the luminal side where substrates were captured (Fig. 2a, c). The chalk-shaped EM density, in conjunction with the results of a liquid chromatography-tandem mass spectrometry (LC-MS/MS) experiment, suggested the presence of a linear endogenous spermine (SPM) substrate (Fig. 2a and Supplementary Fig. 1d, e). Moreover, using an in vitro microscale thermophoresis assay, we found that SPM, as a tetravalent cation, clearly has a higher binding affinity for ATP13A2 over either the trivalent spermidine (SPD) or the divalent putrescine (PUT) (Fig. 2d and Supplementary Fig. 1d). These findings, in agreement with previous studies[7], may partially explain why hATP13A2 preferentially captures SPM.

The residues around Site$_1$, including W251, D254, D463, Q944, F963, and D967, are highly conserved among P5B ATPase members (Fig. 2e and Supplementary Fig. 9). There are also several negatively charged residues, such as D254, E456, and D960 (Fig. 2b, f), located near the entrance to the chalk-shaped pocket, which may be important for the recruitment of positively charged polyamines into Site$_1$. Distinct from the E2P map, the E1-ATP map has no corresponding SPM

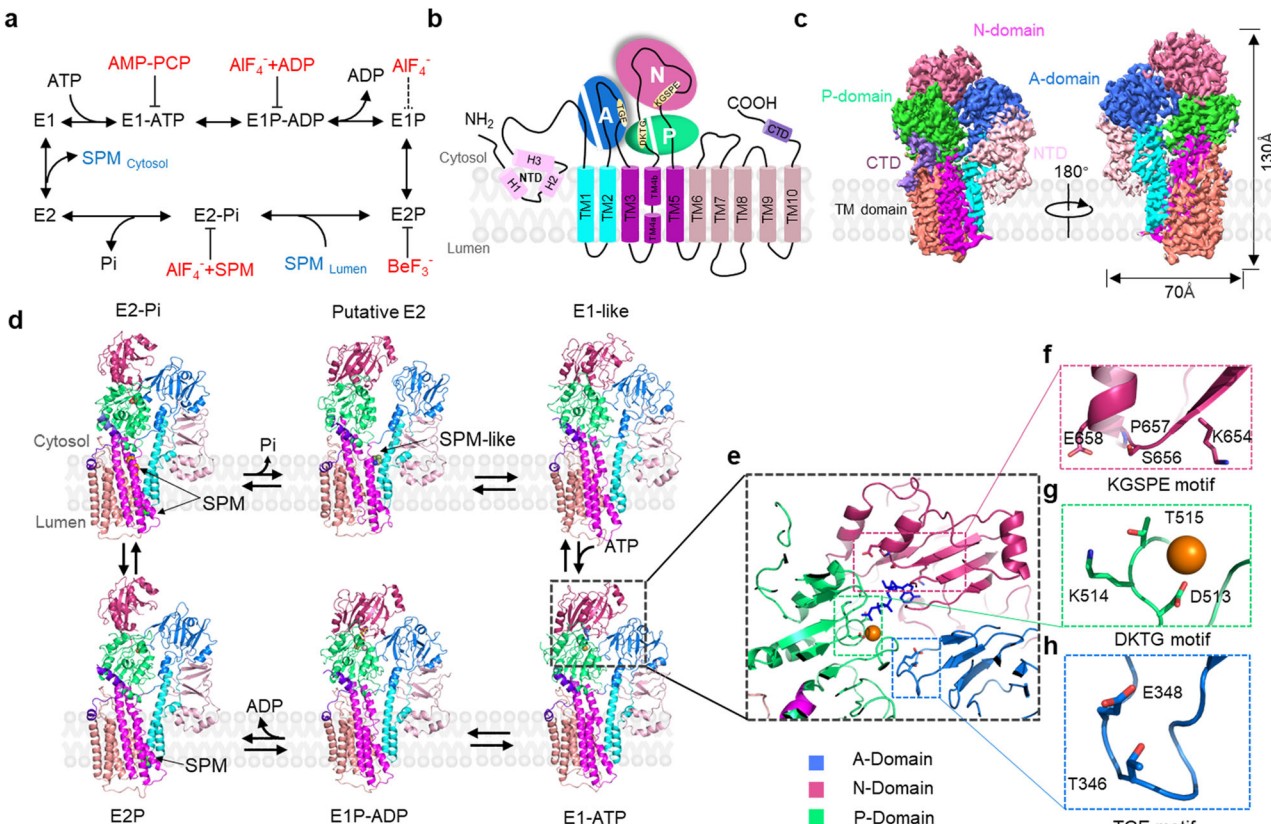

**Fig. 1 | Cryo-EM analysis of human ATP13A2. a** Schematic diagram of Post-Albers cycle for ATP13A2. The transport of SPM is shown in blue. Components that mimic the enzyme intermediates of their respective reaction cycles are shown in red. SPM (marked in blue) is recruited from the lumen side of ATP13A2 in the E2P state and released into the cytoplasm in the E2 state. **b** Topology diagram of hATP13A2. Conserved domains and TM helices are schematically illustrated. In the cytoplasmic regions, the A, N, and P domains and the C-terminal regulatory domain (CTD) are colored marine, pale violet red, limegreen, and medium purple, respectively. TM1-TM2, TM3-TM5, and TM6-TM10 of ATP13A2 are cyan, magenta, and salmon. The specific N-terminal domain (NTD) is colored in pink. The DKTG motif in the P-domain, the KGSPE motif in the N-domain, the TGE motif in the A domain, and TM4

is divided into TM4a and TM4b. **c** Cryo-EM density map of hATP13A2 in the E1-ATP state. **d** Atomic models of hATP13A2 in six intermediate states along the Post-Albers transporting cycle: E1-like, E1-ATP, E1P-ADP, E2P, E2-Pi, and putative E2. The SPM is shown as spheres models. **e** The AMP-PCP molecule tightly bound to the catalytic center in the E1-ATP state. The AMP-PCP nucleotide is shown in blue sticks, and $Mg^{2+}$ ions are shown as orange spheres. The distribution of conserved motifs on the A, N, and P domains are marked by dashed boxes in marine, pale violet red, and limegreen, respectively. **f–h** The key signature motifs KGSPE for ATP coordination within N-domain (**f**), DKTG for auto-phosphorylation within P-domain (**g**), TGE for dephosphorylation within A-domain (**h**). Key residues are represented as colored sticks with labels. The same color scheme is used throughout the manuscript.

density near $Site_1$ (Fig. 2g, h). The further structural comparison suggests that $Site_1$ in the E1-ATP state is covered by three hydrophobic residues—W251, Y256, and F963—which may sterically hinder the binding of polyamine substrates (Fig. 2i). The gating feature of $Site_1$ was further supported by all-atom explicit solvent MD simulations of hATP13A2 in the E2P state, which revealed that, despite being deeply buried in the membrane bilayer, $Site_1$ could be hydrated (Supplementary Fig. 12a). The solvent exposure of $Site_1$ is essential for the recruitment of polyamines from the luminal side and facilitates the hydrophilic substrate binding pathway, reminiscent of the E2P state of the P4-type flippase in which the entry site was also found to be fully hydrated to recruit the hydrophilic head group of its phospholipid substrates[29].

**Polyamine movement**

Phosphorylated P-type ATPases transit from the E2P state to the E2-Pi state, accompanied by aspartic acid dephosphorylation and substrate movement along the transport pathway[17]. However, the mechanism of polyamine movement along the transport pathway after entering $Site_1$ remains elusive. By adding exogenous SPM into the E2-Pi cryo-EM samples, we captured substrate binding at multiple putative binding sites along the transport pathway, which was further confirmed using LC-MS/MS analyses (Supplementary Fig.13a). We found that hATP13A2

in the E2-Pi state could bind more SPM compared to other intermediate states, such as E1P-ADP, E2P, and E1-like (Supplementary Fig. 13b, c), suggesting additional substrate binding sites in hATP13A2.

Solving the structure of hATP13A2 in the E2-Pi state revealed a large inward-opening cavity on the cytosolic side, formed by TM3, TM4, and TM5 (Fig. 3a, b). Inside the cavity, a clear chalk-like EM density was located parallel to TM3 (Fig. 3a) and above the conserved PPALP motif in TM4 (Supplementary Fig. 9). Moreover, the amino acid analysis revealed the large inward-opening cavity is formed by more than a dozen residues, including Y240, F419, S425, F428, V429, L432, V469, P470, M477, L499, N502, and K506 (Fig. 3c), that are highly conserved among the P5B-type ATPase subgroup (Supplementary Fig. 9), indicating that P5B members may share a similar transport mechanism. Next, we used cross-linking mass spectrometry (XL-MS) to confirm potential binding sites for SPM (Supplementary Data 1) and found that SPM could be covalently cross-linked with K506 and K420, which are located at the inward-opening cavity (Fig. 3d and Supplementary Fig. 14). These findings suggest the inward-opening cavity to be the second substrate binding site of hATP13A2 (denoted as $Site_2$). This observation was further supported by our coarse-grained MD simulations, in which no lipid was found in the area, disproving that the observed density was lipid-derived (see next section; Supplementary Fig. 15a–d). To further validate the functional role of $Site_2$, we mutated

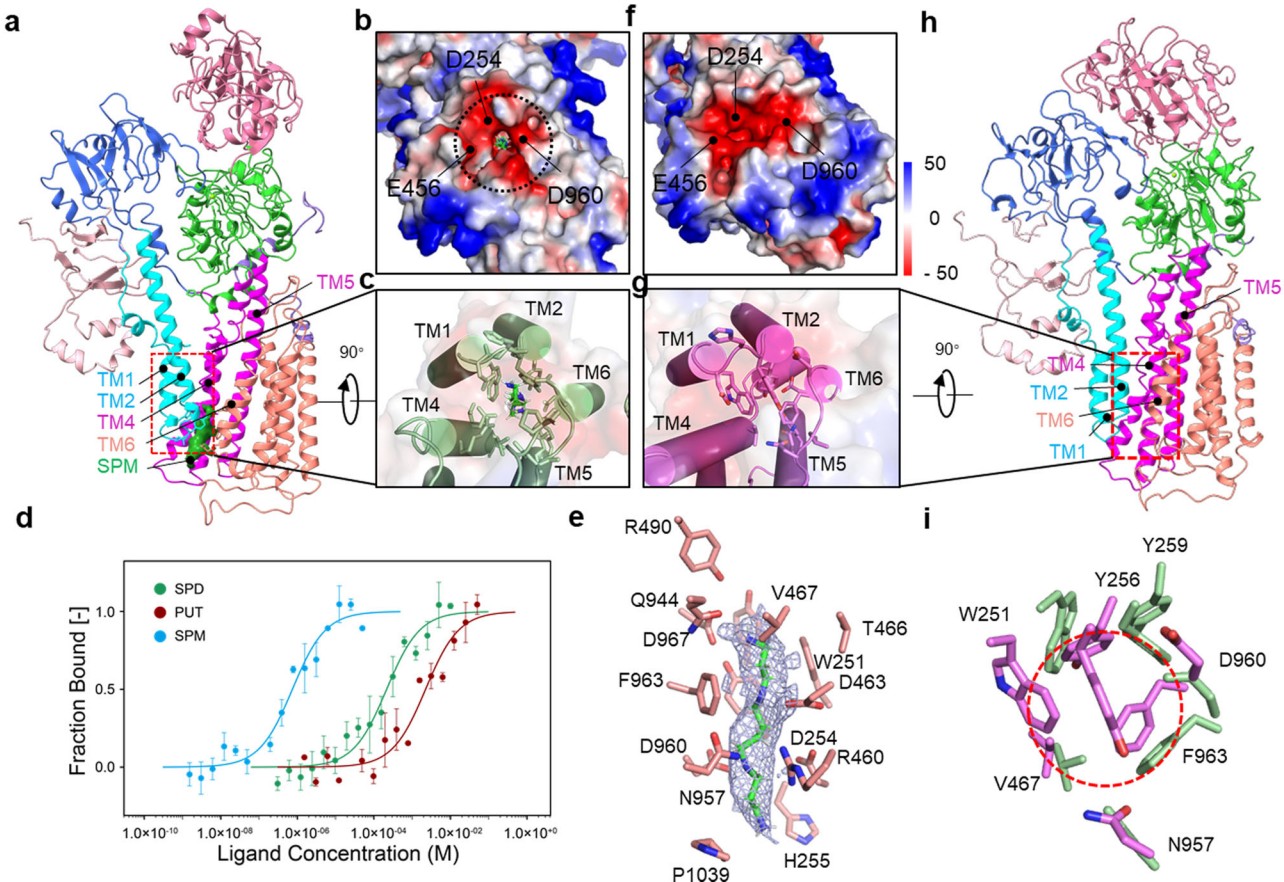

**Fig. 2 | Analysis of the entrance site within hATP13A2. a, h** The atomic model of hATP13A2 in the E2P state (**a**) and E1-ATP state (**h**), respectively. The EM density of SPM in the E2P state is shown as ChimeraX's "solid" (green) representation (**a**). **b, f** Electrostatic potential surface of Site$_1$ within E2P (**b**) and E1-ATP states (**f**), respectively, viewed from the luminal side. The substrate-binding cavity is outlined with a dashed line. The negatively charged amino acids D254, E456, and D960 are displayed at the entrance site. **c, g** The luminal gate is "opened" in the E2P structure (**c**) and "closed" in the E1-ATP structure (**g**). The SPM molecule is shown as green sticks (**c**). **d** MST assays of the polyamine-binding affinity of

hATP13A2 protein. Data from three independently performed experiments were fitted to the single binding model via the MO. Affinity analysis software version 2.3 (NanoTemper Technologies),error bars represent SD (standard deviation). **e** The structure of SPM at the entrance site. The EM density for the bound SPM is shown in mesh, and residues within 4 Å of the bound SPM are shown as sticks. **i** Conformational changes of the entrance binding pocket in the E1-ATP state (magenta) and E2P states (limegreen) by superimposing the main chain atoms of the luminal side. The entrance of polyamine is closed by three hydrophobic residues W251, Y256, and F963 in the E1-ATP state.

the residues in Site$_2$ and quantified SPM-dependent activities of the mutants (Fig. 3e). We found that the Y240A and F419A mutations disrupted SPM-induced ATPase activity (Fig. 3e and Supplementary Fig. 1f, g), suggesting that Site$_2$ is likely involved in the transport pathway.

**Polyamine release**

Structures of hATP13A2 in the E2 state are essential for exploring the mechanism of substrate release. However, yet no E2 structure has been solved for P5A and P5B ATPases. A previous study showed that wild-type ATP13A2 accumulates in cells as an autophosphorylated (E1P or E2P) form in the absence of polyamines[27]. However, supplementation of polyamines in the culture medium can sufficiently accelerate the dephosphorylation of ATP13A2[7,27] to facilitate trapping of hATP13A2 in the dephosphorylated E1 or E2 state. Therefore, to capture its dephosphorylated structure, we pre-incubated hATP13A2 with SPM, in the absence of exogenous ATP or phosphate analogs, and obtained an EM map of dephosphorylated hATP13A2 at 4.8-Å resolution (Fig. 4a and Supplementary Fig. 8).

We consider the dephosphorylated conformation a putative E2 state, which has never been captured for P5B ATPases. In this E2 structure, we observed a significant movement of the A domain, in which the conserved TGE motif was not clearly visible, probably due to

its flexibility (Supplementary Fig. 8h), and separation of TM2 and TM4b, resulting in an open pocket exposed to the cytosolic side that could be used for polyamine release (Fig. 4a–c). Of note, we also observed an apparent elongated density in the release site which we modeled as an SPM molecule (Fig. 4d, e). The release site was found to be highly hydrated in all-atom MD simulations, which suggested several potential release pathways (Supplementary Fig. 12b–d and Supplementary Movie 1).

Around the release site, there are clear unassigned densities commonly observed in most cryo-EM hATP13A2 maps (Fig. 4d, e and Supplementary Fig. 16a–e), which are likely attributable to negatively charged phospholipids interacting with the positively charged surface near the release site (Supplementary Fig. 16f–h). To further investigate the interactions and dynamics of lipids and SPM, we performed coarse-grained MD simulations (10 repeats, each for 10 μs) of the E2 state in a modeled lysosomal membrane[22]. We found that PIP2 clearly out-competed other negative-charged phospholipids (except the abundant POPA) and clustered near the release site in the simulations (Fig. 4f and Supplementary Figs. 12e–h and 15e, f). As a result, negatively charged phospholipids, especially PIP2, interact with the positively charged surface near the release site of hATP13A2 and are likely attributable to the previously unassigned densities (Supplementary Fig. 16).

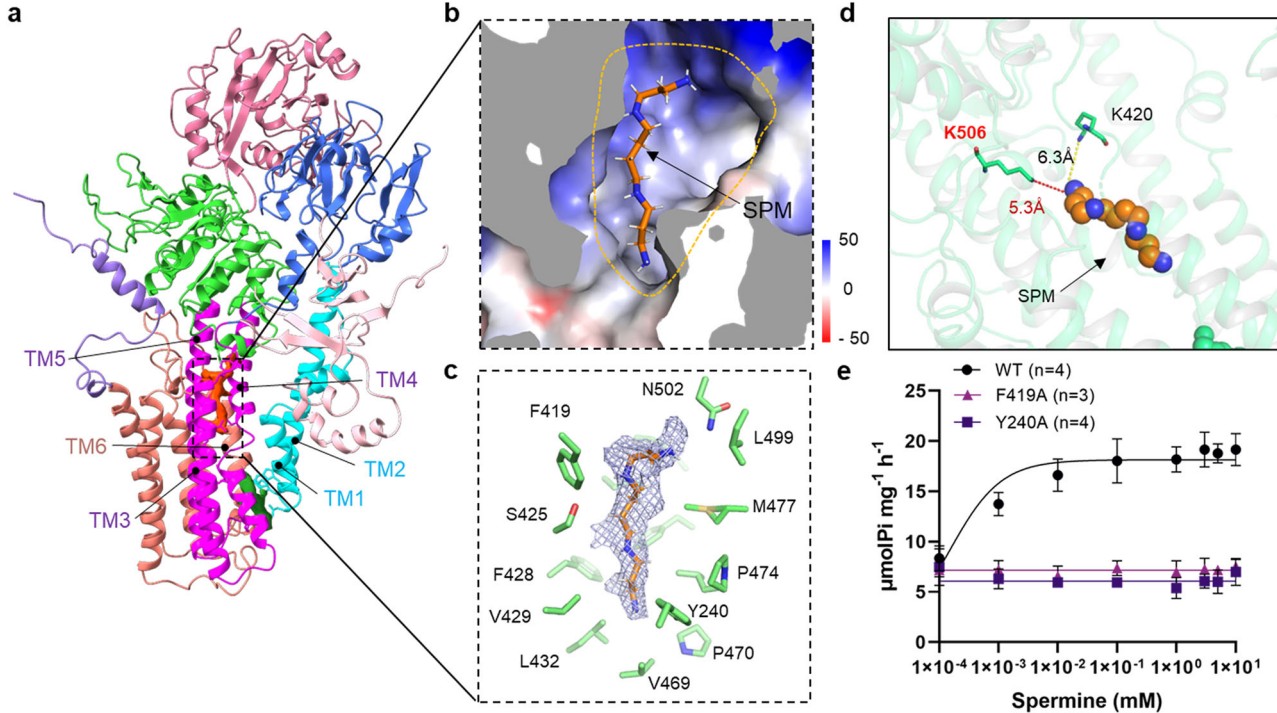

**Fig. 3 | Structural analysis of hATP13A2 in E2-Pi state. a** The overall structure of ATP13A2 in the E2-Pi state. The EM density of SPM is shown as ChimeraX's "solid" (orange) representation at Site$_2$. **b** Electrostatic potential surface of the inward-open cavity within the E2-Pi structure. TM2 and TM4b falling apart makes a large cavity. The SPM molecule is shown as sticks. The yellow-dotted line represents the extent of the SPM binding pocket. **c** Representation of the SPM binding pocket of the E2-Pi conformation at Site$_2$. The cryo-EM density for the bound SPM is shown in mesh, and residues contacting SPM are shown as sticks. **d** Potential cross-linking sites for BS$^3$ around Site$_2$. The dashed lines show the distance between Lysine (K) and SPM. K506 and K420 covalently cross-linked with the SPM. **e** Dose–response curves showing the SPM-induced ATPase activity of purified WT hATP13A2 or mutants. Data points represent the mean ± SD of three (F419A) or four (WT and Y240A) measurements. Lines are fitting by nonlinear regression of the Michaelis–Menten equation. Source data are provided as a Source Data file.

## Conformational rearrangements

To elucidate the conformational changes of hATP13A2 during a complete substrate transport process, we focused on the E1-ATP, the E2-Pi, and the putative E2 states, which have shown distinct conformational rearrangements (Fig. 5). The ATP analog tethers together the N, P, and A domains of hATP13A2 in the E1-ATP state (Figs. 1e–h and 5a). In the post-hydrolysis E2-Pi state, the N-domain rotates outward by ~25° (Fig. 5b), while the release of Pi induces expansion of the cytosolic domains in the following E2 state (Fig. 5c and Supplementary Fig. 11d). In addition to the conformational rearrangements of the cytosolic domains, the putative transport pathway formed by the TM regions were altered. The first conformational change within the TM regions involves Site$_1$, which is closed in the E1-ATP state, open in the E2P state, and closed in the putative E2 state (Supplementary Fig. 11e), indicating a gate function of the entrance site during substrate recruitment. Adjustment of the inward-opening cavity was induced by the separation of TM2 and TM4b (Fig. 5d–f), which formed angles of ~30° and 37° in the E1-ATP and E2-Pi states, respectively (Fig. 5d, e). However, due to dephosphorylation and phosphate release, the A domain rotates away from the P-domain, such that the corresponding angle in the putative E2 structure is about 47° (Fig. 5f), creating ample space between TM2 and TM4b, and resulting in a more solvent-exposed inward-opening cavity that could facilitate the release of SPM. These conformational changes in the E1-ATP, E2-Pi, and putative E2 states suggest that the angle between TM2 and TM4b plays a role in substrate transport.

Due to TM2 and TM4b moving further apart, the phospholipid counterparts in the E2-Pi structure protruded into the cleft between the TM2 and TM4b helices, bringing them close to SPM (Supplementary Fig. 16c, d). Further movement of TM2 and TM4b in the following putative E2 state allowed the counterpart phospholipids to directly interact with SPM (Fig. 4d, e and Supplementary Fig. 16h), which may

affect the release of SPM via an unknown mechanism. This may partly explain why negatively charged phospholipids, such as PIP2, increase the transport activity of hATP13A2[7].

Taken together, the conformational rearrangements of hATP13A2 in the E1-ATP, E2-Pi, and putative E2 states provide a structural basis to fully understand the mechanism underlying hATP13A2 transportation of substrates.

## Proposed transport mechanism

In this study, we solved the structures of six hATP13A2 intermediate states along a nearly complete transport cycle, thereby identifying two distinct substrate binding sites and one release site (Fig. 1a, d). Volume analysis showed that hATP13A2 exhibits a cytosol-facing, solvent-exposed cavity that is substantially larger than those in other P-type ATPases, such as P5A (Supplementary Fig. 12i, j). Surprisingly, this inward-opening cavity was ubiquitous in all states and could even capture polyamine substrates in its binding sites in some intermediate states (E1-ATP, E1P-ADP, E2P, E2-Pi, and putative E2). This finding indicates that, unlike other P-type ATPases, hATP13A2 may transport substrates via a unique mechanism featuring an inward-opening cavity as a substrate buffering tank.

We integrated biochemical assays, structural determinations, and multiscale MD simulations to develop a model of the hATP13A2 transport process (Fig. 6 and Supplementary Movie 2). During the transition from the E1-like to E1-ATP state, the A domain, P-domain, and N-domain of hATP13A2 are tightly tethered together by ATP to form a catalytic center for ATP hydrolysis and aspartic acid phosphorylation (Fig. 1e–h). Then, in the following E1P-ADP–to–E2P transition, ADP release from the catalytic center would induce the outward movement of the N-domain by ~25°, which would subsequently alter the architecture of the TM regions, causing entrance pocket Site$_1$ to open

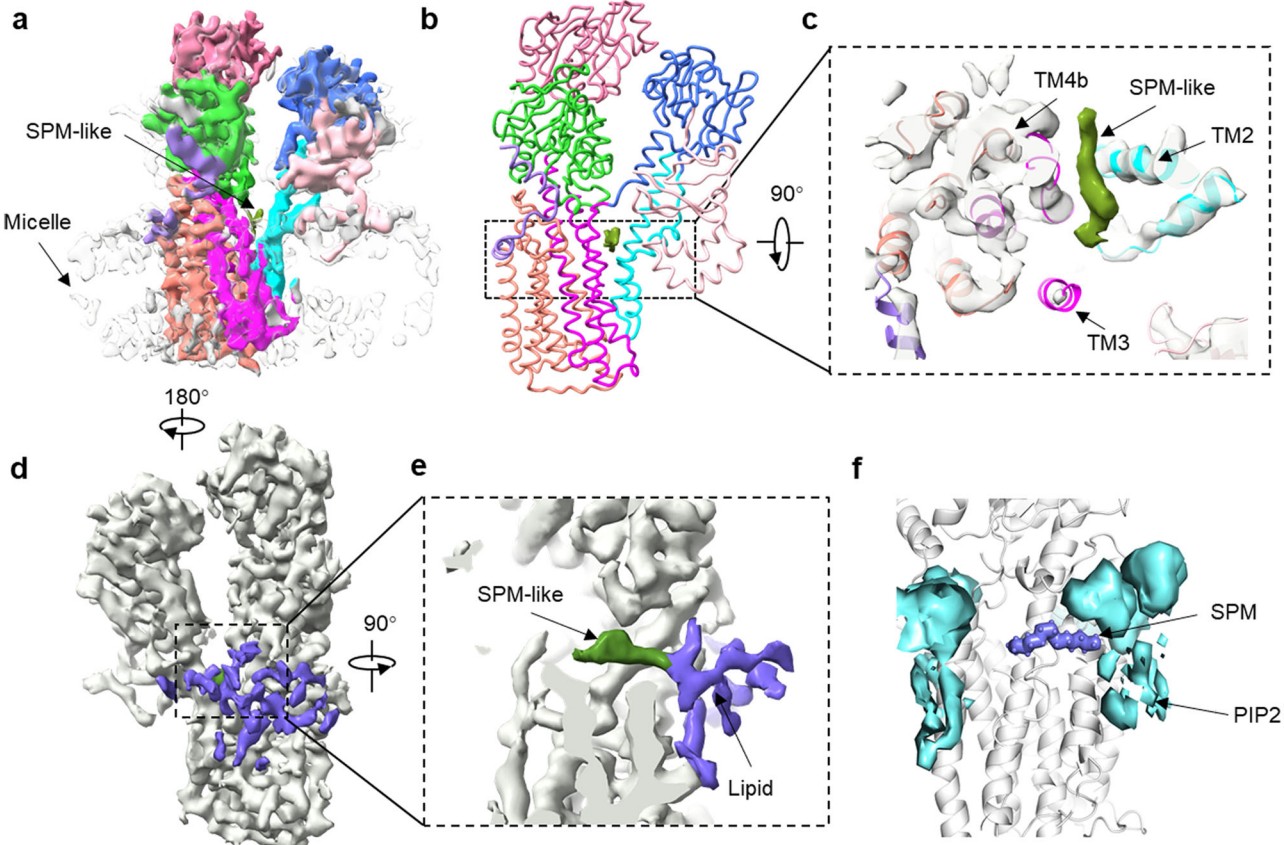

**Fig. 4 | Structural analysis of hATP13A2 in the putative E2 state. a** Cryo-EM density representation of polyamine released. SPM-like density is shown with smudge. The micelle density is shown as gray. **b**, **c** The EM density of SPM in the putative E2 state is shown as ChimeraX's "solid" (smudge) representation (**b**), and a 90° rotation of the structure reveals the putative substrate-releasing pathway between TM2 and TM4b (**c**). **d** Interaction of negatively charged lipids (purple) at the releasing site with positively charged SPM. **e** EM density of the lipid band bound to the transmembrane domain in the putative E2 state. **f** The average density map of PIP2 around the SPM in releasing site obtained from CG MD simulations.

toward the luminal side of the lysosome and allowing the recruitment of a polyamine, after which the polyamine would move deeper into Site$_1$ in the subsequent E2-Pi state. In the transition from E2-Pi to putative E2, Pi release would induce TM region rearrangement to close the outward-opening Site$_1$ pocket and drive the movement of the polyamine substrate from Site$_1$ to the inward-opening cavity, serving as a substrate buffering tank containing Site$_2$ and the release site. The PPALP motif connecting TM4a and TM4b in this step acts as a gating switch between the cytosol and the lysosome lumen, similar to the transport mechanism of Spf1[18], thereby preventing polyamine back release and defining the transport direction of ATP13A2.

Following a typical Post-Albers cycle, the substrate would be transported from the luminal side to the cytosolic side in the transition from E2-Pi to E2. However, unlike in the previous model, polyamine substrates may be released by hATP13A2 in a later stage, depending on the affinity of the substrates for the binding sites and their local concentration. The always-open cytosolic cavity could even accommodate two to three polyamines in some intermediate states (such as E1-ATP, E2P, and E2-Pi), depending on substrate concentration.

The pumping of polyamines from the luminal side to the cytosolic cavity by ATP hydrolysis temporarily raises the polyamine concentration inside the buffering tank. Polyamines follow the concentration gradient and diffuse to the cytosolic side with a minimal energy cost. The following separation of TM2 and TM4b in the E2 state would make the polyamines in the buffering tank move to the release site for release. Meanwhile, phospholipid molecules within their own monolayer near TM2 and TM4b may help remove the polyamines from hATP3A2 in the E2 state.

## Discussion

In this project, we solved the cryo-EM structures of hATP13A2, including six conformational intermediates that cover the complete transport cycle. Four intermediate states (including E1-ATP, E1P-ADP, E2P, and E2-Pi) were solved at relatively high resolutions (3.2–3.7 Å), whereas the E1-like and putative E2 states were solved at medium resolutions (5.6 Å and 4.8 Å, respectively) probably due to the rapid dynamic equilibrium between the E1 and E2 states[35,36]. The two recently reported nominal E1 structures (one wild-type[21] and one mutant[22]) present almost identical conformations that are distinct from our E1 structure (Supplementary Figs. 10 and 11b). We also note that the E1-ATP and E1P-ADP structures we captured have essentially identical conformations, similar to other P-type ATPases[32,34] (Supplementary Fig. 10).

To capture the E1P state, Sim et al. incubated hATP13A2 with phosphate analog AlF$_4^-$ before vitrification[22]. However, using the same buffer conditions, the structure we captured is very similar to the E2P state (RMSD = 0.82 Å) (Supplementary Figs. 10 and 11c) that differed from the Sim's E1P-like structure (PDB: 7N77, RMSD = 6.22 Å) (Supplementary Figs. 10 and 11g, h). This might be partially explained by (1) Sim et al. preparing the E1P-like structure using mutant ATP13A2 (D458N/D962N) whereas ours was based on the wild-type protein, and (2) their use of an insect cell-derived protein to capture the E1P-like structure and our use of a recombinant protein isolated from mammalian cells. The semi-quantitative MS analysis of SPM-bound hATP13A2 in its intermediate states showed that the E2-Pi state could contain more than three times as much SPM as the E2P state, indicating that multiple substrate binding sites are present in addition to Site$_1$ and

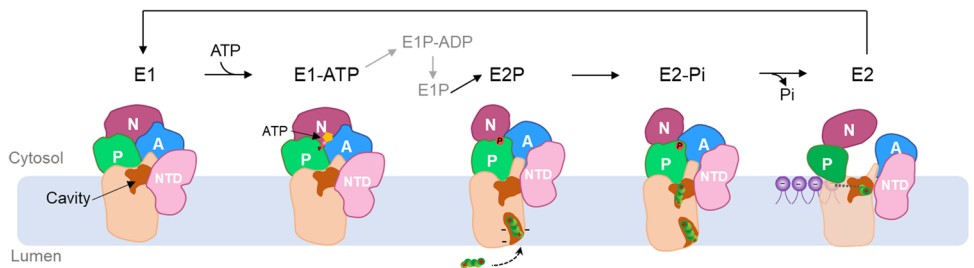

**Fig. 5 | Conformational arrangements among intermediate states.**
**a**–**c** Conformational comparison of A, P, and N Domains in the E1-ATP (**a**), E2-Pi (**b**), and putative E2 states (**c**). Arrows indicate major domain movements between the compared structures. **d**–**f** Cartoon representation of the split movement between TM2 and TM4b from the front view and the luminal view of E1-ATP (**d**), E2-Pi (**e**), and putative E2 states (**f**), respectively. The counterpart angle is marked by the yellow-dotted line. The distances between the ends of TM2 and TM4b are labeled.

**Fig. 6 | A proposed model of hATP13A2 transporting polyamines.** During the E1-to-E1P transition, ATP binding induces the A, P, and N domains to tight together, producing a catalytic center. In E2P state, conformational rearrangements in the cytosolic domains and transmembrane regions induce the opening of the entrance pocket Site$_1$, and the negatively charged amino acids are enabled to recruit positively charged polyamines (shown in green). The Site$_1$ captured polyamines will then move to the inward-open cavity formed by Site$_2$ in the E2-Pi state, resulting in an increasing concentration of polyamines in the buffering tank (shown in brown). Finally, polyamines will diffuse into the cytoplasm through the cleft between TM2 and TM4b in the E2 state. The lipids (shown in purple) attached to the exit might accelerate the release of polyamine into the cytoplasm.

Site$_2$. Interestingly, a C-shaped density was observed near the release site in the E2-Pi state. This is consistent with an SPM molecule binding prior to release, although further verification is required.

The P1, P2, and P3 ATPases are ion pumps whose substrates are small and very soluble, so that they might be released immediately once transported to the release site. The P4 and P5A ATPases translocate lipids and transmembrane helices, respectively, which need to be captured in a larger cavity. The ATP13A2 has a unique inward-facing cavity, composed of Site$_2$ and the release site, with a large volume, which provides an opportunity for binding an extra SPM molecule. The low pH environment of the lysosome may favor the recruitment of polyamines from the luminal side to the negatively

charged entrance Site$_1$ in the E2P state. Characterizing the subsequent movement of the polyamine substrate from the entrance binding pocket to the inward-opening cavity on the cytosolic side in the following E2 state provides valuable insight into how polyamines are released from hATP13A2. We captured the structure of the putative E2 state of a P5B-type ATPase[20–23]. Structural comparison revealed that the putative E2 state of ATP13A2 differs from that of other P-type ATPases[37,38], with distinct conformational rearrangements (Fig. 5). This structural information further reveals the transport mechanism of hATP13A2.

Dysfunction of ATP13A2 influences the homeostasis of $Zn^{2+}$, possibly by disrupting the fusion process between the lysosome and autophagosome[39]. Overexpression of ATP13A2 promotes the fusion of the lysosome and autophagosome through HDAC6[25]. However, the molecular mechanism of ATP13A2-mediated regulation of the fusion process is still unclear. Lipids may play an important structural and functional role in membrane proteins as suggested by the crystal and cryo-EM structures in which the densities of sparse lipids have been occasionally captured[40,41]. Accordingly, a belt of substantial densities near the substrate release site was clearly captured in our EM maps and further identified as PIP2 or POPA by our coarse-grained MD simulations, consistent with previous and recent reports[22,42]. This finding implies that these phospholipids actively regulate hATP13A2 transport.

More than 20 mutations in hATP13A2 have been reported to be related to neural diseases such as KRS and SPG78[8–11]. Some of these mutations may cause disease by influencing the transport function of hATP13A2. We found that these mutations were distributed widely across different regions rather than located in a single domain (Supplementary Fig. 16i), indicating that they may impact the activity of hATP13A2 via different mechanisms.

In summary, we reported the structures of a nearly complete conformational cycle of a human polyamine transporter and proposed a substrate transport mechanism, which we believe the mechanistic findings will be valuable in guiding rational drug design, even if we have no exact strategies at this moment to use these findings.

## Methods

### Plasmids
The full-length human ATP13A2 (NM_022089.3) gene was cloned into the pCAG vector containing a C-terminus tandem twin Strep-tag by homologous recombination using the primers A13-F, A13-R (Supplementary Table 4). Its point mutants were generated with the Fast Mutagenesis System kit (Transgen, FM111) with primer pairs in Supplementary Table 4, and sequences were verified by Sanger DNA sequencing (RuiBiotech, Beijing).

### Protein expression and purification
The hATP13A2 was expressed with the human embryonic kidney (HEK) 293-F cell line using polyethylenimine (PEI) (Polysciences, Inc.) transient transfection method according to the manufacturer's instructions. Briefly, just before cell transfection, cells were seeded to give the desired cell density of between $2.0 \times 10^6$ cells/mL and $2.5 \times 10^6$ cells/mL. For transfection, ~1.5 mg of the hATP13A2 plasmid were premixed in 40 mL of the serum-free medium before 3 mg PEI reagent was added, then incubated at room temperature for 15–30 min. The PEI/DNA mixture was added to 800 mL cells and incubated at 37 °C in the presence of 5% $CO_2$. After 16 h, 10 mM sodium butyrate was added, and the temperature was reduced to 30 °C for 72 hours before harvesting. Cells were harvested by centrifugation ($1500 \times g$, 10 min, 4 °C), frozen in liquid nitrogen, and stored at −80 °C until use. For purification, cells were thawed and gently resuspended in lysis buffer for 2 h containing 50 mM MOPS (pH 7.0), 100 mM KCl, 10% glycerol, 5 mM $MgCl_2$, 1 mM dithiothreitol (DTT), 1% (wt/vol) n-dodecyl-β-d-maltopyranoside (DDM, Anatrace), 0.2% (wt/vol) cholesteryl hemisuccinate (CHS, Sigma) with EDTA-free protease inhibitor cocktail (Roche). All

subsequent steps were performed at 4 °C. The solubilized fraction was clarified by centrifugation at $51,428 \times g$ for 60 min, and it was incubated with Strep-Tactin® Sepharose® (IBA, LifeSciences) for 1.5 h by rotation. After incubation, the resin was collected and washed with 20 column volumes of wash buffer containing 50 mM MOPS (pH 7.0), 100 mM KCl, 10% glycerin, 5 mM $MgCl_2$, 1 mM DTT, 0.05% (wt/vol) DDM, 0.01% (wt/vol) CHS, 1 mM DTT and 1 mM phenylmethylsulfonyl fluoride (PMSF). Finally, the ATP13A2 was eluted with 50 mM MOPS (pH 7.0), 100 mM KCl, 10% glycerin, 5 mM $MgCl_2$, 0.025% (wt/vol) DDM, 0.005% (wt/vol) CHS, 50 mM biotin. The eluate was concentrated and further purified by size-exclusion chromatography (SEC) on a Superose™ 6 Increase column (GE Healthcare) in a SEC buffer containing 50 mM MOPS (pH 7.0), 100 mM KCl, 5 mM $MgCl_2$, 1 mM DTT, 0.06% digitonin. Peak fractions were collected and concentrated to ~5–10 mg/mL before cryo-EM grid preparation.

### ATPase activity assay
ATPase activity was assessed using a commercially available kit by measuring the release of inorganic phosphate (Pi) from ATP according to the kit protocol (Nanjing Jiancheng Bioengineering Institute, China). Briefly, Enzymatic reactions were performed for 10 min (37 °C) in a final volume of 68 μL containing 2 mM ATP. The purified WT or mutants ATP13A2 (0.2–0.5 μg) in buffer containing 50 mM MOPS (pH 7.0), 100 mM KCl, 10 mM $MgCl_2$, 1 mM DTT, 0.03% (wt/vol) DDM, 0.006% (wt/vol) CHS, various concentrations of the SPM were incubated at 37 °C for 10 min. The reactions were then mixed with 42 μL of matrix reagent for 10 min at 37 °C and terminated by adding 10 μL of reagent buffer. The supernatant was taken to measure the Pi content after centrifuging at $1200 \times g$ for 10 min. Transfer 30 μL of the supernatant to a 96-well plate and then mix with 100 μL of chromogenic agent for 2 min at room temperature, followed by the addition of 100 μL of Reagent 6 for 5 min. The absorbance was measured at 636 nm using the microplate reader (Tecan, Switzerland). Nonlinear regression to the Michaelis–Menten equation and statistical analysis was performed using GraphPad Prism version 9.0.0.

### Microscale thermophoresis (MST) assay
hATP13A2 were cloned into a pCAG vector and then transformed into HEK293F to express the His-tagged recombinant proteins. The purified recombinant hATP13A2 proteins were diluted to 200 nM in MST assay buffer (50 mM MOPS (pH 7.0), 100 mM KCl, 10 mM $MgCl_2$, 1 mM DTT, 0.03% (wt/vol) DDM and 0.006% (wt/vol) CHS). The diluted proteins were then labeled by dye (RED-tris-NTA) using a Pierce™ BCA Protein Assay Kit (Thermo Fisher Scientific). Next, polyamine solutions were prepared using MST assay buffer and mixed completely with the labeled proteins in various concentrations. The reaction was incubated for 30 min at room temperature in the dark before the samples were analyzed by the Monolith™ NT.115 instrument (NanoTemper Technologies) at 25 °C using the MST data acquisition software according to the manufacturer's instructions. Data from three independently performed experiments were fitted to the single binding model via the MO. Affinity analysis software version 2.3 (NanoTemper Technologies).

### EM sample preparation and data collection
To capture hATP13A2 in different intermediate states, the purified hATP13A2 were mixed with different inhibitors or substrates at the following final concentrations: E1-ATP, 2 mM AMP-PCP; E1P-ADP, 5 mM NaF, 1 mM $AlCl_3$, 5 mM ADP; E1P, 10 mM NaF, and 2 mM $AlCl_3$; E2P, 10 mM NaF, and 2 mM $BeSO_4$; E2-Pi, 10 mM NaF, and 2 mM $AlCl_3$, 2 mM SPM; E2, 2 mM SPM. After incubation for 1–2 h on ice, the mixture was centrifuged at $16,000 \times g$ for 10 min at 4 °C, then 4 μL aliquots of the ATP13A2 sample were applied to a freshly glow-discharged (PELCOeasiGlow, 0.39 mBar air, 15 mA, 50 s) Quantifoil R1.2/1.3 300 mesh Au holey carbon grids (Quantifoil). The grids were blotted for 3–4 s, and then plunge-frozen in liquid ethane using Vitrobot Mark IV

(Thermo Fischer Scientific) operated at 10 °C and 99% humidity. All grids were then transferred and kept in liquid nitrogen for data collection. For E1-like state ATP13A2 (apo), freshly purified protein samples in SEC buffer were applied immediately.

The prepared grids were transferred to a Titan Krios G3i microscope (Thermo Fisher Scientific), running at 300 kV and equipped with a Gatan Quantum-LS Energy Filter (GIF, slit width of 20 eV) and a Gatan K3 Summit direct electron detector in the electron counting mode. Imaging was performed at a nominal magnification of ×81,000, with the super-resolution pixel size at 0.5475 Å each pixel for the E1-like, E1-ATP, E1-ADP nominal E1P, E2P, and E2-Pi state datasets. Dose-fractionated images were recorded at a nominal magnification of ×64,000, with the super-resolution pixel size at 0.54 Å each pixel for the putative E2 state dataset. All movie stacks were collected with the serialEM[43] or EPU software with a defocus range of −0.8 to −1.2 μm. All datasets were collected using image-beam-shift multiple recording model. The total accumulated dose on the specimen was about 50 e⁻/Å² with a total exposure time of 8 s. Each micrograph stack contains 32 frames. The number of micrographs collected for each stated grid is described in Supplementary Table 1.

### Cryo-EM data processing

The single-particle analysis procedures of all states are summarized in Supplementary Figs. 2–8. For all datasets, the dose-fractionated movies were subjected to beam-induced motion correction using MotionCor2[44] software or patch motion correction (cryoSPARC)[33], and the contrast transfer function (CTF) parameters were estimated using CTFFIND4[45] or patch CTF estimation (cryoSPARC)[33]. Micrographs were inspected to remove micrographs that were not suitable for image analysis. All subsequent image processing was performed using cryoSPARC. For all datasets, one hundred micrographs were selected to train the Topaz model using a manual curate exposures job in cryoSPARC. Particles were picked by blob picker and extracted with a box of the size of 300 pixels, Fourier-cropped to 100 pixels, and subjected to rounds 2D classification. Selected particles from 2D classification and the 100 micrographs were then trained for the Topaz model using Topaz train in cryoSPARC. Then, particles of all micrographs were picked by Topaz extract using the trained topaz model in cryoSPARC[46] and were extracted with a box of 300 pixels for all datasets. All particles were used for 2D classification. After 2D classification, particles were selected for the Ab-initio reconstruction to generate 4 initial maps in cryoSPARC. Subsequently, the particles were classified by multi-rounds of heterogeneous refinement using the four initial models. The final particles selected from heterogeneous refinement were subjected to non-uniform refinement and local and global CTF refinements to yield a final map. The overall resolution of the final map was determined by the 0.143 criteria of the gold-standard Fourier shell correlation (FSC)[47]. Local resolution estimation was used to determine the local resolution of each map in cryoSPARC.

### Model building and refinement for cryo-EM structures

For all states, Phyre2 online server[48] was used to get the initial model of ATP13A2 for model building. The initial model of ATP13A2 was fitted to the map, and the coordinates relative to the cryo-EM maps were saved in UCSF ChimeraX[49]. Models for all structures were built after rigid-body fitting of individual domains into the corresponding maps using the initial model and rounds of coordinate adjustment and local refinement in coot[50]. Model refinement was performed using Phenix (phenix.real_space_refine)[51] with secondary structure restraints and with the refinement resolution limit set to the overall resolution of the map. Structural figures were prepared using UCSF ChimeraX[49] and PyMOL (www.pymol.org/). The final refinement statistics are provided in Supplementary Table 1. For the ligands bound ATP13A2 complex, the coordinates of ligands were obtained through the 3-letter code

(code: SPM) and merged into the coordinates of every complex using Coot.

### Parameters and protocols of molecular dynamics simulations

The all-atom systems of putative E2 and E2P states were embedded into a lipid bilayer with 40% POPE:40% POPC:20% cholesterol and solvated with a 0.1 M NaCl aqueous solution, resulting in ~285,000 atoms in total for each system. The dimensions of the equilibrated box were 13.5, 13.5, and 14.0 nm in the $x$, $y$, and $z$ dimension, respectively. The temperature was kept constant at 310 K using the Nose−Hoover thermostat with a 1 ps coupling constant, and the pressure at 1.0 bar using the Parrinello−Rahman barostat with a 5 ps time coupling constant. We used a cutoff of 1.2 nm for the van der Waals interactions using a switch function starting at 1.0 nm. The cutoff for the short-range electrostatic interactions was at 1.2 nm and the long-range electrostatic interactions were calculated by means of the particle mesh Ewald decomposition algorithm with a 0.12-nm mesh spacing. A reciprocal grid of 120 × 120 × 120 cells was used with 4th-order B-spline interpolation. We used the CHARMM36m force field[52] and prepared the systems using the CHARMM-GUI web interface[53]. Force field parameters for the phosphorylated Asp513 were the same as previously described[54], and force field parameters for SPM were generated using the CHARMM General Force Field[55]. To investigate the hydration of the SPM substrates in the binding site, we used Gromacs 2019.6 to perform atomistic MD simulations, 200 ns for each, with SPM and protein restrained in the cryo-EM conformations.

Coarse-grained (CG) MD simulations of the putative E2 and E2-Pi state were performed in a model lysosomal membrane using the Martini 2.2 force field[56] and Gromacs 2020.3 or 2020.6[57]. The coarse-grained Martini structures of systems were prepared using the *martinize.py*[56]. The asymmetric lipid bilayer with compositions, $c$ (cytosolic leaflet) and $l$ (luminal leaflet), was prepared using the script *insane.py*[58]. The $c$ composition consists of 15.2% cholesterol, 19.8% POPC, 32.0% POPE, 23.0% POPA, 3.0% POPS, 3.5% POPI, and 3.5% PIP2 (c16:0/18:1 phosphatidilinositol 3-4 bi-phosphate), and 38.8% cholesterol, 46.2% POPC, and 15.0% POSM in the luminal leaflet (Supplementary Table 2). The system was subsequently solvated with CG water box, 10% of which were replaced by anti-freeze (WF) Martini water beads.

In the CG simulation, we maintained a temperature of 310 K with the v-rescale thermostat[59]. After well equilibrating the systems in the NVT and NPT, production runs were carried out in the NPT ensemble for 10 μs using the Berendsen barostat[60]. In both simulations of the putative E2 and E2-Pi state, hATP13A2 was restrained in the cryo-EM conformations. For putative E2, the SPM molecule observed in cryo-EM density was included and restrained. While for E2-Pi, to predict the SPM binding sites, the SPM molecules in cryo-EM model were not included in the simulations. Ten independent simulations were performed for each system to collect statistical data. The last 5 μs trajectories were used for their interactions with SPM and the protein. To calculate the average density map, we used the last 2 μs of each trajectory from ten simulations. The lipid density map was analyzed with GROmaps[61]. The volume analysis of the cytosolic cavities was performed using the Computed Atlas of Surface Topography of proteins (CASTp) method[62].

### Liquid chromatography with tandem mass spectrometry (LC-MS/MS) analysis

To perform the semi-quantitative analysis of the SPM bound by ATP13A2 in different intermediate states, including apo E1-like, E1P-ADP, E2P, and E2-Pi, divided the purified hATP13A2 protein into four equal parts and mixed with the corresponding inhibitor, respectively (Refer to EM sample preparation). After 1 h of incubation on ice, the proteins were subjected to SEC with the SEC buffer containing the corresponding inhibitor, respectively. Peak fractions were collected

and quantified by Bicinchoninic acid (BCA) assay (Beyotime Inst). The same amount of protein sample was used to prepare the sample for mass spectrometric semi-quantitative analysis. Briefly, hATP13A2 proteins were extracted with acetonitrile containing 0.1% formic acid with a volume ratio of 1:4. The reaction system was then centrifuged at 16,000×g for 15 min, and the supernatant was transferred to a new tube for dryness through evaporation with nitrogen. After that, the sample was re-dissolved with 100 μL of Milli-Q water containing 0.1% formic acid, and centrifuged at 16,000×g for 10 min at 4 °C. Finally, the supernatant was saved for LC-MS/MS analysis.

LC-MS/MS analysis was carried out on a Thermo LC-MS Q Exactive-Orbitrap system with a Kinetex 1.7 μm C18 100 Å LC column (100 × 2.1 mm, Phenomenex). Milli-Q water containing 0.1% formic acid (solvent A) and acetonitrile (solvent B) were used as mobile phases, and all solvents used were LC-MS grade. The flow rate was set at 0.2 mL/min. A 100 μL injection volume was used for the analysis. The capillary temperature was set at 320 °C, and aux gas heater temperature was set at 370 °C. The solvent gradient was as follows (solvent B%): 1–1% (0–1.5 min), 1–5% (1.5–4.5 min), 5–100% (4.5-5.5 min), and 100–1% (5.5–7.0 min). The MS data were acquired in positive ions modes.

## Cross-linking mass spectrometry (XL-MS) analysis

The purified hATP13A2 (1 μg/μL) were cross-linked with BS³ (bis[sulfosuccinimidyl] suberate, Sigma) in a 1:1 (w/w) ratio at room temperature for 30 min. The reaction was terminated by adding 20 mM ammonium bicarbonate. Run the cross-linker sample on SDS-PAGE to examine the cross-linking reaction. The Coomassie Brilliant Blue stained gel band was excised and digested with trypsin (Promega) for 16 h at 37 °C. The solution was dried out and desalted using a Mono-Spin C18 column (GL Science, Tokyo, Japan). The peptide mixture was analyzed by a customized 30-cm-long pulled-tip analytical column (75 μm ID packed with ReproSil-Pur C18-AQ 1.9-μm resin, Dr. Maisch GmbH); the column was then placed in line with an Easy-nLC 1200 nano HPLC (Thermo Scientific) for mass spectrometry analysis. The analytical column temperature was set at 55 °C during the experiments. The mobile phase and elution gradient used for peptide separation were as follows: 0.1% formic acid in water as buffer A and 0.1% formic acid in 80% acetonitrile as buffer B, 0–1 min, 3–6% B; 1–96 min, 6–36% B; 96–107 min, 36–60% B, 107–108 min, 60–100% B, 108–120 min, 100% B.

Data-dependent MS/MS analysis was performed with a Q Exactive-Orbitrap mass spectrometer (Thermo Scientific). Peptides eluted from the LC column were directly electro-sprayed into the mass spectrometer with the application of a distal 2.5-kV spray voltage. A cycle of one full-scan MS spectrum (m/z 355–2000) was acquired, followed by top 20 MS/MS events, sequentially generated on the first to the twentieth most intense ions selected from the full MS spectrum at a 28% normalized collision energy. Full-scan resolution was set to 70,000 with automated gain control (AGC) target of 3e6. MS/MS scan resolution was set to 17,500 with an isolation window of 1.8 m/z and AGC target of 5e6. The number of microscans was one for both MS and MS/MS scans, and the maximum ion injection time was 20 and 50 ms, respectively. The dynamic exclusion settings used were as follows: charge exclusion, 1 and >8; exclude isotopes, on; and exclusion duration, 40 seconds, respectively. MS scan functions and LC solvent gradients were controlled by the Xcalibur data system (Thermo Scientific).

## Identification of cross-links with pLink 2

The search parameters used for pLink 2 were as follows: instrument, HCD; precursor mass tolerance, 20 ppm; fragment mass tolerance 20 ppm; cross-linker BS³ + SPM(340.2838) (cross-linking site lysine, only considering mono-link, BS³ + SPM mass shift 340.2838). Cysteine alkylation by iodoacetamide was specified as fixed modification and Methionine oxidation, protein N-terminal acetylation as a variable;

peptide length, minimum 6 amino acids and maximum 60 amino acids per chain; peptide mass, minimum 600 and maximum 6000 Da per chain; enzyme, trypsin, with up to three missed cleavage sites per cross-link. Protein sequences of ATP13A2 proteins were used for database searching. Precursor and fragment tolerances were 10 and 20 ppm, respectively. The results were filtered by requiring a spectral false identification rate ≤0.01. MS2 spectra were annotated using pLabel[63], requiring mass deviation ≤20 ppm.

## Reporting summary

Further information on research design is available in the Nature Portfolio Reporting Summary linked to this article.

## Data availability

The data that support this study are available from the corresponding authors upon reasonable request. The cryo-EM maps and the coordinates in this study have been deposited in the Electron Microscopy Data Bank (EMDB) and Protein Data Bank (PDB) under accession codes: EMD-35384 and 8IEK for E1-ATP state; EMD-35385 and 8IEL for E1-like state; EMD-35392 and 8IES for E1P-ADP state; EMD-35386 and 8IEM for E2P state; EMD-35387 and 8IEN for E2-Pi state; EMD-35388 and 8IEO for nominal E1P state; EMD-35391 and 8IER for putative E2 state. Mass spectrometry proteomics data have been deposited to ProteomeXchange Consortium (http://proteomecentral.proteomexchange.org) via the PRIDE partner repository with the dataset identifier PXD037493. Source data are provided with this paper.

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

## Acknowledgements

We thank all staff members of the Cryo-EM Centre, Southern University of Science and Technology and Chunlong Guo, Zhenqian Guo, Chuan Liu, Fanhao Meng, Li, and other staff members at Shuimu BioSciences Ltd. for their assistance in data collection. We thank Dr. Chao Peng and Weiqian Wang of the Mass Spectrometry System at the National Facility for Protein Science in Shanghai (NFPS), Shanghai Advanced Research Institute, Chinese Academy of Science, China for data collection and analysis. We acknowledge Desheng Liu, Jia Wang's help in model building. We thank Hongwei Wang for his careful reading of this work. This work was supported by funds from the National Science Foundation of China (Grant 32000850 to Z.M.L.), Shenzhen Municipal Basic Research projects (Grant JCYJ20210324105007020 to Z.M.L.), the start-up funds of Southern University of Science and Technology to Z.M.L., the National Key R&D Program of China (2021YFF1200404 to Y.W. and 2021YFA1201200 to R.H.Z.), the National Center of Technology Innovation for Biopharmaceuticals (NCTIB2022HS02010 to R.H.Z.), the Fundamental Research Funds for the Central Universities (K20220228 to Y.W.) and the Information Technology Center and State Key Lab of Computer-Aided Design (CAD) & Computer Graphics (CG), Zhejiang University. This work was also partially supported by the National Independent Innovation Demonstration Zone Shanghai Zhangjiang Major Projects (ZJZX2020014 to R.H.Z.), Shanghai Artificial Intelligence Lab (P22KN00272 to R.H.Z.), the Starry Night Science Fund of Zhejiang University Shanghai Institute for Advanced Study (SN-ZJU-SIAS-003 to R.H.Z.).

## Author contributions

Z.L. and Y.W. initiated and supervised the project. J.M., M.W., Y.H., and R.B. prepared and purified the proteins. C.X. and K.L. prepared the cryo-EM specimens. B.Y., J.H., Q.J., X.C., and J.M. collected the EM data. J.M. and C.X. performed the cryo-EM analysis and the model building. Z.L., Y.W., R.Z., and K.C.C. performed model refinement. Y.W. and L.F. performed the MD simulations. Z.Y. and A.H. performed the LC/MS assay. M.N. and H.L. performed the XL-MS analysis. Z.L. and Y.W. drafted the manuscript with help from all authors.

## Competing interests

The authors declare no competing interests.
