## [Peer Review File · Nature Communications]

Conformational cycle of human polyamine transporter ATP13A2REVIEWER COMMENTS

Reviewer #1 (Remarks to the Author):

ATP13A2 is a polyamine transporter in the lysosomal membrane that plays an important role in maintaining the homeostasis of polyamine concentration. ATP13A2 mutations have been found connected to several hereditary human diseases. This manuscript by Mu et al reports seven structures of human ATP13A2 determined with single particle cryo-EM. These structures illustrate a nearly complete picture of the transport process in ATP13A2. It is noted that six of these seven structures are similar to the previous structures of human ATP13A2 published recently. In addition to the known structures, here the authors report the structure of ATP13A2 in the E2 state, which is new and adds the last piece of structures to fill the gap in the ATP13A2 transport cycle. Besides the new structure, this study also reveals two previously unknown substrate binding sites located in a large cavity exposed to the cytosolic side. Based on the structural interpretations and evidence from ATPase activity assays and MD simulations, the authors propose a transport mechanism that utilizes the cytosolic cavity as the buffering tank for substrate release, which is different from the other well-characterized P-type ATPases. Although some conclusions remain speculative, this study will make a significant contribution to understanding the polyamine transport mechanism.

Major comments:

1. There must be tremendous efforts to obtain these seven structures of human ATP13A2 protein. An important part missing in the current manuscript is a comparison with those recently published cryo-EM structures of human ATP13A2. It will be very useful to put forward the new discoveries/interpretations the authors have had from their own structures that have counterparts published by the other groups. It makes sense that the authors put the structures in the story line from polyamine entry, movement, to release, but it is not every clear to me what has been known and what is new when those known structures are not mentioned.
2. The authors incubated ATP13A2 protein with SPM to obtain the claimed E2 state. The details of how this sample preparation was done are not provided. No further experiments were performed to further characterize it or to confirm it is corresponding to the long-sought-after E2 state. Meanwhile, the resolution of the cryo-EM structure in this state is poor. A speculation is that the sample was in highly flexible or heterogeneous states where the protein might be falling apart. The atomic model built at this resolution (4.9 Å) is not reliable, either. I would strongly suggest that the authors carry out experiments to prove it is the E2 state and improve the resolution of this structure. Otherwise, it is more appropriate to claim “a putative E2 state”.
3. I do not think 4.9 Å resolution allows identifying the density of SPM or lipids in the cryo-EM map. The MD simulations may provide some support to the interpretations, but those densities could be from noise or heterogeneous structures of detergent or something else. Without experimental data, such as using a (modified) substrate with a feature easily recognizable at low resolution, it is too risky to claim a substrate in the density map. In addition, are densities of SPM observed in both Site2 and Site3?

4. Is a channel exiting from the transmembrane domain to the cytosolic cavity observed in the E2 state? This may serve a strong support to the state in which the substrate is being released.
5. What conformational changes happen in the transmembrane domain during the transition from the E2-Pi state to the E2 state? This is a critical step of transport when the substrate moves from the luminal side to the cytosolic side. There may not be sufficient structural information to explain this process, but it will be very interesting to speculate and discuss possible mechanisms for this action.
6. The authors discovered two new substrate binding sites (Site2 and Site3) in the E2-Pi state when SPM was added to the sample. This seems plausible but again is speculative until solid evidence is available. Did the densities of SPM on these two sites disappear when no SPM was added to the E2-Pi sample? The manuscript shows that mutating some residues in these two sites reduced the SPM-induced ATPase activity (Fig. 3b). But it is unclear whether this reduced activity is SPM-dependent or not. A plot of activities at a series of SPM concentrations will help visualize this.
7. Since the cytosolic cavity is exposed to the solvent in all the states reported in the manuscript, can the binding of SPM to Site2 and Site3 be observed in the other states besides the E2-Pi? If there are 3 binding sites, is it supported by the MST assays or other binding affinity assays?
8. There are several obvious errors in the manuscript, e.g., Fig. S4 is a duplicate of Fig. S3. I am concerned that there are also a few less obvious errors that the reviewers are unable to catch. I ask the authors do a thorough job to correct errors coming from manuscript preparation. Some figure captions are not well written.
9. The methods do not include sufficient details about how individual experiments were carried out.

Minor points:

1. A number of abbreviations are not spelled out when they first appear in the manuscript.
2. The term “electron density” is misused several times. The maps from cryo-EM reconstructions are not electron density maps.
3. Fig. 1f: should labels M1-10 be TM1-10?
4. Fig. 1h: are labels TM4 and TM5 swapped?
5. The term “stereo view” is used in the supplementary figure captions multiple times, but there are no stereo views.
6. Line 136-137: “A blob of density corresponding to AMP-PCP (an ATP analog) was well solved near the KGxPE motif in the N domain (Fig. 1b)”. But the actual Fig. 1b does not show this.
7. Line 274 and 275: what are “-1e” and “-5e”?
8. Line 650-651: “the atomic model ribbon is shown (grey), the protein depicted in ribbon representation (gray)”. It is unclear what this means.

Reviewer #2 (Remarks to the Author):

- This manuscript by Mu et al. provides highly significant data and interesting interpretations for the field of the P-ATPases transporters particularly in the P5B subgroup which was shown to transport polyamines. This work advances the knowledge of the ATP13A2 which has been implicated in several neurological diseases.
-
- The manuscript focus on a hot area that is the structure of the protein ATP13A2. Other studies have been recently published on the same topic: Sim et al., Mol Cell, 2121;- Tillinghast et al., Mol Cell,2021; Chen et al., Cell Disc. 2021; Tomita et al., Mol. Cell 2021. These studies use essentially the same methodology that is the CryoEM structural analysis of the human ATP13A2, and together with the paper of Li et al. Nat. Commun. 2021 Jun 25, describing the structure of the yeast homolog yPK9, they provide a wealth of data on the structural features of the P5B-ATPases.
-
- Given the new data and the novel mechanisms proposed in my opinion this manuscript would require A) a more detailed description of the experimental conditions, B) a better explanation of the proposed interpretation of the data, C) a comparison with the results provided by recent studies. Each point is summarized below.

Major points

1. P16 line 469. Please indicate the composition of the ATPase reaction media, including the concentrations of Mg²⁺ and ATP. What is the enzyme specific activity in Fig1 Supp. For example in umol/mg.min. What is the meaning of about 50% ATPase of the WT when no SPM is added? How is this incorporated in the Albers-Post scheme?
2. P16 line 497. Please describe the experimental media including ligands and protein concentration in the MST assays. How these media compare to the CryoEM?
3. P17 line 507. Please provide a detailed composition of the media used for the each CryoEM structure, indicating concentration of protein, lipids, ions, nucleotides, nucleotide analogs, Pi analogs etc., incubation time and conditions before data acquisition.
4. P7 line 192. In addition to the SPM site 1 which was described in other studies, this manuscript describes novel extra binding pockets at site 2 and site 3 in the E2.Pi and E2 structures. This is a quite interesting observation. The E2 structure presented here, although at low resolution, is totally novel. However, the E2.Pi structure has been described in other studies and in contrast with the authors' findings, only SPM site1 was reported. In this regard, A) What action led the authors to see the other sites? B) Is there a P-ATPase in which the substrate has been observed in "transit" with several binding sites in the same structure? Please explain and add to Discussion.
5. Fig 3 shows a model of the protein in E2 structure with three SPM molecules. It can be interpreted as if the protein could bind three SPM molecules simultaneously. In contrast I noticed that the MST curves were fitted to single binding site model. A) What is the stoichiometry SPM/AT13A2 in E2? B) If E2P-E2

species have SPM bound in different sites there should be a difference in the binding energy. What is the fraction of protein with SPM bound at each site? Also, is difficult to see how this large substrate would move through the protein without parallel changes in the protein.

6. P10 line 303. The authors propose a mechanism of stimulation of ATP13A2 activity by PIP2 which is very different from that previously proposed for yPK9 by Li et al. Nat. Commun. 2021 Jun 25. A) Please discuss. B) How PIP2 affects SPM affinity? In the model proposed by the authors PIP2 accelerates Koff of SPM. Can the MTS experiment be done in the presence and in the absence of PIP2? Alternatively, how the dependence of ATPase activity with SPM is affected by PIP2?

7. P12 line 371. The structural similarity in the E1P and E2P reported by the authors is at variance of previous studies of other P-ATPases. Furthermore, in the model presented in P26 line 691 the authors state “In the following transition of E1P to E2P, ADP release causes conformational rearrangements in the cytosolic domains and transmembrane regions, opening the entrance pocket ...”. Please explain.

8. P11 line 353. The concept of “buffering tank” is very appealing. However I am not aware of a similar proposal in other P-ATPase. Why ATP13A2 would be different in this respect? Could this be related to the SPM gradient between lysosome and cytosol and the direction of the transport?

- Minor points

1. P16 line 471. “Follow the mixture”. Please rephrase

2. P5 lines 109-112. Perhaps is more adequate to indicate that this is true for P4 and P5 ATPases only and not for all P-ATPases.

3. P6 line 166. Perhaps the authors mean that “among P5B-ATPases members and not in P5A-ATPases.

4. P7 line 179 “solvent explosion “? Do you mean exposure?

5. Figure 1. Could you please add the transported substrate in the transport cycle?

6. Figure 2. Legend. “An illustrator””? Do you mean an illustration?

Reviewer #3 (Remarks to the Author):

The paper “Structural basis of a full conformational cycle of human ATP13A2 transporting polyamines” describes a polyamine transport pathway from lumen to cytosol based on multiple cryo-EM structures. Several structures are presented that confirm the overall architecture and polyamine entry on the luminal side and overall, the data appear solid. However, four papers (see below) from different labs have already reported multiple conformations of the same transporter studied here, ATP13A2, and as such the novelty of the data can be questioned. Further underscoring this, the authors refrain from doing comparisons to any of these previous studies. The authors have also not presented what is still missing in the field, following these previous achievements. This is important as it will highlight the potential significance of the current findings – what new information is the new data bringing to the

field? Throughout the paper, it is not clear how the claimed conformations have been stabilized and assigned in the transport cycle. Are the states inward or outward-facing, are the TM domains open or not, is substrate present or not and does the location change, which is intimately coupled to the E1-E1P-E2P-E2 cycle, or if they are similar to each other or to already available structures, which is relevant regarding the novelty. This is important also because many of the states reported in the recent papers are almost indistinguishable (even if different inhibitors have been supplemented), and hence they add little new information to regarding the biological or biochemical function of the protein. It is possible or perhaps even likely that such a scenario is also the case here, but I do not know as comparisons are missing. The authors own summary of their results in Fig. 5 suggests five different intermediates, but the abstract indicate that seven intermediates have been identified. The authors claim that they have uncovered a previously uncharacterized E2 state of P5B-ATPase, with an inward-open cavity, also be present in other states, with two bound spermine molecules, despite the low overall resolution of that particular state renders modelling challenging. Also, inward-facing conformations are typically linked to E1 (inward-open) or E1P (inward-occluded) states in P-type ATPases and hence the authors likely need to revise how they assigned the new state. Combined with molecular dynamics simulations it is suggested that the surrounding phospholipids can interact with the polyamine to stimulate its release, which is an interesting model. It is also recommended that the paper is sent for language review before resubmission, as there are a large number of errors in the text. One such example is that it should be “transport pathway” not “transporting pathway”, which consistently incorrect throughout the text.

Specifications of the above-mentioned remarks:

*The four previous multi-conformational structural studies on ATP13A2 (<https://pubmed.ncbi.nlm.nih.gov/34728622/>, <https://pubmed.ncbi.nlm.nih.gov/34715014/>, <https://pubmed.ncbi.nlm.nih.gov/34715013/>), <https://pubmed.ncbi.nlm.nih.gov/34798056/>) and the related <https://pubmed.ncbi.nlm.nih.gov/34172751/> should be referenced and the findings summarized in the introduction. What are the missing gaps in the field?

*It needs to be made clearer how do you mimic or stabilize the multiple states you have determined, and in the text, it must be made clearer which state the structure being referred to in any specific part of text is in, including what stabilizer is used. Details of which stabilizers were used, how and in which concentrations should be clarified in the methods. Currently there is one row (507) covering this, which is insufficient. Clarify and explain each configuration (e.g., inward open or outward open)? How do you distinguish these states? Different states mean different configurations (e.g., open/occluded) instead of relying on the different additives. Is the inward open cavity present in all states or a specific E2 state? If it is always open to the cytosol, how would polyamine back release be prevented?

* The manuscript lacks comparisons, both with other available P5B structures, but also between the different structures in the current manuscript. How different are the different structures you have determined? How does the spermine compare between them? How does all this relate to previously determined structures? For example, it appears that you have two structures with spermine in site3, but

there is no information on if the conformation of spermine is different in the two structures. What are the conformational changes between these states?

I also have the following minor points:

Introduction

Row 52: It should be “Kufor-Rakeb”, not “Kufor-Rekab”

Row 53: What is SPG78? Why have you explained what Kufor-Rakeb is, but not this one?

Row 60: How is Mn²⁺ homeostasis related to ATP13A2 and polyamine transport?

Row 65: Include more details about the different states. How are these intermediate states stabilized? What is different between each state, which ones are opened toward lumen or cytosol and which are closed, for example? It would be a benefit for people who are not familiar with P-type ATPases.

Row 68: This sentence makes it sound like P5s are unique among P-types because they are only found in eukaryotes and they have subfamilies, which is incorrect.

Row 73 (and elsewhere): It appears that P1-P5 are referred to as subfamilies, but P5A-P5B are also referred to as subfamilies. Please use different terms for them to avoid confusion.

Row 75: What was presented in these papers needs to be expanded upon. Also, what does it mean that they “were online”? They have been published.

Rows 79-86: This paragraph should be expanded. Describe the N-terminal region, as it has not been introduced before? What is known about the interaction with lipids? How do they interact?

Rows 98-99: Is it possible to understand the pathogenic mechanism only based on the structure and functional analysis of ATP13A2? How is drug development relevant to the current paper, as small molecules rarely can increase function?

Results

Row 105-106: Please comment on your high levels of ATP hydrolysis in the absence of added cargo.

Overall structure section: considering the overall structure of hATP13A2 has been described several times before, this is not a novel finding and could be considerably condensed. You may consider moving parts of it to the introduction.

Row 156 (and row 258, among others): It occasionally sounds like ATP13A2 has a channel structure instead of a transporter. For example, what do you mean by a “channel-shaped architecture” in row 156? Or that “several channels were observed to connect to site3 in row 258? Because it is not a channel.

Row 156: What does “where substrates dominate” refer to? That there is substrate bound in the entry site? Or that the concentrations of polyamines is higher in the lumen than cytosol? But why would it then need a transporter?

Row 165 and figure S2: ATP13A1 is a P5A ATPase, not P5B, which explains the poor sequence homology!

Row 190-192: What does this sentence mean? I feel particularly confused about “would subject SPM to encompass the hATP13A2 extensively”.

Polyamine movement section: What is function of site2? Does it exist in other P5B-type ATPases?

Row 247: How does this compare to the other structure with spermine in site3?

Row 255-277: Why are these negatively charged phospholipids detected by MD close to site3, but not site2?

Row 262: This makes it sound like the water-filled cavity is new for this structure, but have you not already described it?

Discussion

Rows 367-372: This belongs in results.

Methods

Row 434: Here it is stated that a catalytically dead mutant is generated, but it is not referenced anywhere in the text. There is also no information in the methods of how the other mutants were generated, and specifically which mutants they were.

Row 440: Just before transfection, the cells were transfected?

Row 491: Just at the start of the row “1% to% (0-1.5 min). What is the second %?”

Row 525: It is stated that more than 3500 movies were collected for each structure, but based on Table 1 that is not true for E1P-ADP and E1-ATP.

Row 530: Figures S5 to S11 are reference. Should it not be figures S3 to S9?

Figures

The following figures are never referenced, and should be removed or be referenced to: 1h, 2a and 2c.

The supplementary figures do not appear in the order that they are first referenced in the text. Based on the text, the supplementary figures should have the following order: S1, S3-9, S2, S11-12, S10 and S13.

The following figures need their color schemes clarified/adjusted: 1a should state what the color scheme is in the legend: which colors are used for which parts? In fig 3a the spermines themselves are colored, not be pockets. Also, spermine in site3 is not magenta. S2 states that cyan is the least

conserved, but all cyan regions are completely conserved. In S10a the side chains appear to have different colors than the helices they are part of, which does not seem to be the case in S10c. How is the surface colored in S10e and f? In S11d and S12b there are no empty histograms, which the legend states.

Fig 1h: Why is this part of a main figure?

Fig 2a: Why is this necessary?

Figs 2j and S1: The text is too small to be readable in all (2j) or parts of (S1) these figures.

Fig 6: Why does this figure have 5 models, when you claim to present 7 different states? Presenting E2 in two rotated forms is unnecessary in this kind of scheme.

Fig S4h: There is no figure S4h.

Fig S10h: Why is this part of figure S10? Would it not be more appropriate in figure S1?

Fig S12a: Which state(s) did you use to generate these?

Reviewer #4 (Remarks to the Author):

Jianqiang Mu et al present a structural study of the human ATP13A2 polyamine transporter. By using cryo-EM they solve the structure of this transmembrane protein in seven different conformational states. The cryo-EM structures are accompanied by molecular dynamics simulations and functional studies. All together, the study allows to rationalize the functional cycle of this protein, including the steps in which it undergoes major conformational rearrangements to enable the access and egress of polyamine molecules, and remarkably the putative involvement of lipids.

I find this is an extremely interesting study which substantially advances our understanding of the polyamine transport across lysosome membranes. The possibility to capture the protein under different conditions provides a complete structural picture of this important process.

I have the following comments

- It is unclear to me how the (un)binding of ATP, ADP and Pi at the cytosolic units drives the seen conformational rearrangements at the transmembrane side. The manuscript largely focuses on the structural rearrangements at the transmembrane segments, which is of course relevant as there is where the polyamine molecules enter and exit. However the paper does not mention in sufficient detail how these changes connect to the ATP (and ADP+Pi)-dependent conformational changes of the A,N, and P cytosolic units. As the study nicely solved the structure of the whole protein, and in different states, it will be extremely interesting to check the connection between what is happening at the catalytic sites and the accessibility of the transmembrane sites.

In particular, I wonder to what extent the here-determined structures explain how the release of ADP opens the (distant) site 1 to enable polyamine entrance and how the release of Pi connects sites 1 with 2 and 3 to facilitate polyamine passage.

- The authors used MD simulations to check if the S2 and S3 sites could be potentially filled by lipid molecules. They did not observe that. The observation that negatively-charged phospholipids specifically bind to the protein to promote substrate release could have been elaborated further with this technique. The egress of polyamine molecules from the sites via the cytosolic lipid leaflet, relevant during the transport cycle, could have also been monitored in the simulations to support the lipid-mediated polyamine-release hypothesis.

- The Authors indicate that structures were available for other members of this ATPase family and even for this protein in few of the states. A comparison with these previous structures will help to realize conserved-structural features across the family.

- lines 261-263: The mentioned water-filled cavity can not be seen in the Figure S11h. Maybe a zoom to the region of interest will help here.

Minor:

- Fig.s 2a and 5a not mentioned in the text. The scheme in Fig 6 is very useful and could be used through out all the figures to highlight the states that are being presented in the context of the whole transporting cycle.

- Sub figures are not ordered according to their mentions in the text (example: figures 2de-g are presented first than fig 2b-c). Reading flow will benefit of figures being presented in ordering number.

- Abstract and line 51: "hATP13A2". "h" of human? has not been defined before.

- Line 65: what do E1 and E2 stand for?

- Fig.s 1f-h may work better as Fig.s 1a-c to help readers to understand the overall protein topology, before entering into the details of the structures and the cycle.

Response to referees' comments

We thank the referees for their valuable time in reviewing our manuscript and their constructive suggestions. We have carefully taken these comments into consideration and prepared a more thorough and clear revision. We hope our revision addresses them all. In particular, We've added detailed comparison with recently published structures, and MS experiments to provide more evidence to support the presence of extra substrate binding sites in the cytosolic cavity. Please find below a point-by-point response to the referees with our responses in **Blue** and the referees' comments in **Red**.

Reviewer #1 (Remarks to the Author):

ATP13A2 is a polyamine transporter in the lysosomal membrane that plays an important role in maintaining the homeostasis of polyamine concentration. ATP13A2 mutations have been found connected to several hereditary human diseases. This manuscript by Mu et al reports seven structures of human ATP13A2 determined with single particle cryo-EM. These structures illustrate a nearly complete picture of the transport process in ATP13A2. It is noted that six of these seven structures are similar to the previous structures of human ATP13A2 published recently. In addition to the known structures, here the authors report the structure of ATP13A2 in the E2 state, which is new and adds the last piece of structures to fill the gap in the ATP13A2 transport cycle. Besides the new structure, this study also reveals two previously unknown substrate binding sites located in a large cavity exposed to the cytosolic side. Based on the structural interpretations and evidence from ATPase activity assays and MD simulations, the authors propose a transport mechanism that utilizes the cytosolic cavity as the buffering tank for substrate release, which is different from the other well-characterized P-type ATPases. Although some conclusions remain speculative, this study will make a significant contribution to understanding the polyamine transport mechanism.

Major comments:

1. There must be tremendous efforts to obtain these seven structures of human ATP13A2 protein. An important part missing in the current manuscript is a comparison with those recently published cryo-EM structures of human ATP13A2. It will be very useful to put forward the new discoveries/interpretations the authors have had from their own structures that have counterparts published by the other groups. It makes sense that the authors put the structures in the story line from polyamine entry,

movement, to release, but it is not every clear to me what has been known and what is new when those known structures are not mentioned.

Reply:

We thank the referee for their positive comments and very helpful suggestions for our study. Accordingly, to describe the current knowledge of hATP13A2' structures, we have added some detailed descriptions on the previously published hATP13A2 structures to the introduction section of the revised manuscript (line 66-73). The E1-like structure (displaying a different conformation from published E1 structures) and the only solved putative E2 structure provided the knowledge of two intermediate states unsolved in the previous works. Moreover, the new information deduced from structural comparisons between our structures and these published ones was also added to the result and discussion section (Fig. R1 and R2) (line 126, 130, 140, 320, 324, 327, and 329). We hope that the revised manuscript would be more readable.

Fig. R1 (Now Supplementary Fig. 10) Structural comparison of our structures and recently solved structures of ATP13A2 and homologous proteins in different intermediate states.

Fig. R2 (Now Supplementary Fig. 11) Comparison analysis of hATP13A2 structures. **a, b** The structure of hATP13A2 in the E1-like state (cyan) is compared with that of the E1-ATP state (brown) (a), and the D458N/D962N mutant in the E1-apo state (PDB: 7N76, purple) (b), respectively. Arrows are used to indicate the direction of movement. **c, g** The structure of hATP13A2 in the AlF_4^- bound nominal E1P state (pink) is compared with that of the BeF_3^- stabilized E2P state (yellow) (c), and the structure of D458N/D962N mutant in the AlF_4^- bound E1P-like state (PDB:7N77, blue) (g), respectively. **d** Structure of hATP13A2 in the E2-Pi state (salmon) is compared with that of the putative E2 state (green). Zoomed-in view showing conformational changes in TM2 and TM4b helices upon E2-Pi to putative E2 transition. **e** Comparison of transmembrane regions of polyamine entry sites in E1-ATP, E2P and putative E2 state, respectively. **f** The structure of hATP13A2 in the E1-like state (cyan) is compared with the structure of the apo form of Spf1 (PDB: 6XMP, red) (f). **h** As in g, structural changes of the A, N, and P domains are indicated by the distances of the KGSPE motif, DKTG motif, and TGE motif, respectively. The conserved motif of cytoplasmic domain in the nominal E1P state is colored in red, and the 7N77 is colored in yellow.

2. The authors incubated ATP13A2 protein with SPM to obtain the claimed E2 state. The details of how this sample preparation was done are not provided. No further experiments were performed to further characterize it or to confirm it is corresponding to the long-sought-after E2 state. Meanwhile, the resolution of the cryo-EM structure

in this state is poor. A speculation is that the sample was in highly flexible or heterogeneous states where the protein might be falling apart. The atomic model built at this resolution (4.9 Å) is not reliable, either. I would strongly suggest that the authors carry out experiments to prove it is the E2 state and improve the resolution of this structure. Otherwise, it is more appropriate to claim “a putative E2 state”.

Reply:

We thank the referee for their valuable suggestions. We have added more details on the sample preparation for this putative E2 state in the Materials and Methods section, as follow: “... E2, 2 mM spermine. After incubation for 1 ~2 h on ice ...”(line 443-447). We agree that more experimental data or high-resolution structure of E2 state would be useful to understand the polyamine release mechanism. We had indeed managed to collect a substantial amount of data (>10,000 movies) so as to improve the resolution of the E2 state, but only resulting in the medium resolution.

The hATP13A2 sample was prepared in a buffer containing 2 mM SPM. Despite the resolution of the structure is not high enough, we do believe the structure was captured in a E2 state based on two reasons:

1) Veen et al. reported the addition of SPM clearly stimulated dephosphorylation of hATP13A2 (Fig. 1c and d in Nature volume 578, pages419-424 (2020)), which promotes the accumulation of hATP13A2 in the E1 or E2 state¹. Thereby, to capture hATP13A2 in the dephosphorylated state, the protein in SEC buffer was incubated with 2 mM SPM on ice for 1 h before plunge freezing. Under such conditions, the hATP13A2 proteins were mostly in a dephosphorylated state, which indicates the final EM map obtained would probably be assigned to the E1 or E2 state.

2) The structural comparison to other states also supports that this structure was very likely in the E2 state rather than the E1 state. In this structure, the TM2 and TM4b fall apart, creating a release site outward-facing the solution side (Fig. 4) (line 217-219) which is a key feature of the E2 state. The outward-facing feature was further supported by MD simulations and release pathway analysis (Fig. R2d).

But just for safe, we tuned down the statement and used the term of "putative" in the revision.

3. I do not think 4.9 Å resolution allows identifying the density of SPM or lipids in the cryo-EM map. The MD simulations may provide some support to the interpretations, but those densities could be from noise or heterogeneous structures of detergent or something else. Without experimental data, such as using a (modified) substrate with a feature easily recognizable at low resolution, it is too risky to claim a substrate in the

density map. In addition, are densities of SPM observed in both Site2 and Site3?

Reply:

Yes, we did observe densities in both Site2 and Site3 that can fit with SPM molecules, but we also agree with the reviewer that the 4.9 Å resolution is not sufficient to allow us to clearly distinguish SPM and the lipid tails. We indeed have considered using a fluorescent modified SPM to provide further experimental evidence, but did not find commercial products on the market. Therefore, we have to leave this for further studies. We have turned down the statement of Site₃ and considered it as a release site in the revised manuscript. Finally, to make the manuscript more readable, we used the term “SPM-like” to describe the density located at the release site of the putative E2 structure in the revised version of the manuscript (line 684).

4. Is a channel exiting from the transmembrane domain to the cytosolic cavity observed in the E2 state? This may serve a strong support to the state in which the substrate is being released.

Reply:

We agree with him/her that the channel analysis of the structures may provide strong indication of the substrate release pathway. In the putative E2 structure, we did find that the movement of TM2 and TM4b resulted in a large open cavity facing to the cytosolic side which was highly hydrated as observed in our MD simulations (Fig. 5). We speculate that this cavity may be used for the SPM release (Supplementary Fig. 12d). In addition, given that there was SPM-like density in this cavity, we propose that this structure represents a putative E2 state in which the substrate is going to be released (line 223).

5. What conformational changes happen in the transmembrane domain during the transition from the E2-Pi state to the E2 state? This is a critical step of transport when the substrate moves from the luminal side to the cytosolic side. There may not be sufficient structural information to explain this process, but it will be very interesting to speculate and discuss possible mechanisms for this action.

Reply:

In the revision, we have added the description of what conformational changes happen

in the transmembrane domain during the transition from the E2-Pi state to the E2 state (line 258-263 in the Conformational rearrangements subsection). We have also added a figure (Fig. R2d) to show the overall conformational re-arrangement from the E2-Pi state to the putative E2 state and speculate the possible mechanisms for substrate movement in the revised manuscript (line 242). Moreover, we discussed the possible mechanisms of translocation of SPM from the membrane out leaflet to the inner leaflet, and also the mechanism underlying preventing the back release of SPM (line 289-296).

6. The authors discovered two new substrate binding sites (Site2 and Site3) in the E2-Pi state when SPM was added to the sample. This seems plausible but again is speculative until solid evidence is available. Did the densities of SPM on these two sites disappear when no SPM was added to the E2-Pi sample? The manuscript shows that mutating some residues in these two sites reduced the SPM-induced ATPase activity (Fig. 3b). But it is unclear whether this reduced activity is SPM-dependent or not. A plot of activities at a series of SPM concentrations will help visualize this.

Reply:

We thank the referee for their positive comments and very helpful suggestions. We did not observe apparent densities on these two sites in the nominal E1P structure which was prepared with the same additives except SPM. We agreed with the reviewer that in the former version of manuscript, the evidence supporting new substrate binding sites was not very strong.

To provide further support, we have done LC-MS/MS experiment for semi-quantitative analysis of SPM in the E2-Pi state (Fig. R3). We found that hATP13A2 in the E2-Pi state captured more SPM than the other intermediate states, including E1P-ADP, E2P, and E1-like (Fig. R3c), which indicates the existence of extra binding sites for SPM, besides Site₁. Next, we used cross-linking MS to probe the potential binding sites of SPM (Fig. R4), and found that SPM could be covalently crosslinked with K506 (Fig. R4, R5d and e), an amino acid located at Site₂. This finding provides a strong support of Site₂ as a SPM binding site.

Unfortunately, we didn't observe cross-linking of SPM to Site₃, therefore, we have turned down the statement of Site₃ and described it as a release site in the revised manuscript (line 221, 222, 227, 230, 233, 271, 292, 308, 337, 346, and 364).

According to this referee's suggestion, we have added the SPM-dependent ATPase activities of hATP13A2 for the WT and mutants in the revised manuscript (Fig. R6). Mutations in Site₂ showed a decreased SPM-dependent ATPase activity, indicating that

it is probably an SPM binding site.

Fig. R3. (Now Supplementary Fig. 13) Semi-quantitative analysis of SPM bound by hATP13A2 in different intermediate transport states by LC-MS/MS.

a A schematic diagram of the determination of hATP13A2 substrate content in different intermediate transport states by LC-MS/MS (left) and flowchart summarizing the specific process of polyamine content determination (right).

b SDA-PAGE analysis of protein quality.

c Bar graph featuring the abundance values of spermine quantified by mass spectrometry. The graph is performed as the means \pm SD of five independent experiments using GraphPad Prism version 9.0.0. statistical software.

Fig. R4. (Now Supplementary Fig. 14) Probing the potential binding sites for spermine in hATP3A2 by cross-linking mass spectrometry (XL-MS)

a XL-MS analysis workflow for probing hATP13A2 cross-linked spermine.

b Chemical structure of BS³ cross-linking reagent. Spacer arm 11.4 Å. This cross-linker reacts with primary amines (lysine side chain, protein N-terminus).

c Reaction of a cross-linker with a primary amine. The leaving group, part of the cross-linker, is substituted with the primary amine to form a covalent bond between the spacer and the amine. Once the cross-linker had already reacted on its other end, R can stand for either the rest of the cross-linker or another protein or SPM.

d A higher energy collision-induced dissociation (HCD) MS/MS spectrum recorded on the [M+4H]⁴⁺ ion at m/z 717.9042 of the protein hATP13A2 (Q9NQ11) peptide FC(+57.02)IHPLRINLGGK(+340.28)LQLVC(+57.02)FDK harboring one SPM site. Matched ions are labeled in the spectrum and indicate that protein is modified on K506.

e A higher energy collision-induced dissociation (HCD) MS/MS spectrum recorded on the [M+4H]⁴⁺ ion at m/z 671.3622 of the protein hATP13A2 (Q9NQ11) peptide FC(+57.02)IHPLRINLGGK(+154.09)LQLVC(+57.02)FDK harboring one SPM site. Matched ions are labeled in the spectrum and indicate that protein is modified on K506.

Fig. R5. (Now Fig. 3d) Potential cross-linking site for BS³ around Site₂. The dashed lines show the distance between SPM and K506 which could be covalently crosslinked.

Fig. R6. (Now Fig. 3e) Dose-response curves showing the spermine-induced ATPase activity of purified WT hATP13A2 or mutants. Data points represent the mean ± SD of three to four measurements. Lines are fitting by nonlinear regression of the Michaelis-

Menten equation.

7. Since the cytosolic cavity is exposed to the solvent in all the states reported in the manuscript, can the binding of SPM to Site₂ and Site₃ be observed in the other states besides the E₂-Pi? If there are 3 binding sites, is it supported by the MST assays or other binding affinity assays?

Reply:

We thank the referee for their great questions. Although we have not directly observed the binding of SPM to Site₂ in other states besides the E₂-Pi, pieces of evidence supported SPM could be bound in addition to Site₁ in other states. For the E₁-like state of human ATP13A2, we vitrified our purified hATP13A2 protein without adding any SPM, ATP analogs, or phosphate analogs. Structural analysis of our final E₁-like state showed that its structure had a conformation similar to the E₁-ATP state (Fig. R2a), in which Site₁ presents a closed architecture without SPM. However, the semi-quantitative analysis supported that the protein we used to prepare the E₁-like state contains high levels of SPM (Fig. R3c). Moreover, results showed that hATP13A2 in the E₁P-ADP state also contains a high amount of SPM (Fig. R3c). Thereby, all of these led us to speculate that there must be extra binding sites for SPM besides Site₁, very possibly Site₂ and the release site.

According to the referee's suggestion, we also tried to perform the MST assay for stoichiometry; however, using the MST assay to determinate the interaction stoichiometry requires rather high affinities. Stoichiometry experiments are difficult for low-affinity interactions with K_ds in the intermediate or high μ M range since high concentrations are needed for both target and ligand (Cited by: FAQ: Stoichiometry Experiment; <https://nanotemper.my.site.com/explore/s/article/Can-stoichiometry-measurements-be-performed-with-MST>). As a result, the Monolith detector is saturated due to too high fluorescence. Therefore, we used the mass spectrometric analysis to do the semi-quantitative analysis of SPM in different intermediate states.

8. There are several obvious errors in the manuscript, e.g., Fig. S4 is a duplicate of Fig. S3. I am concerned that there are also a few less obvious errors that the reviewers are unable to catch. I ask the authors do a thorough job to correct errors coming from manuscript preparation. Some figure captions are not well written.

Reply:

Thanks for the suggestions. We have gone through the manuscript very carefully and tried our best to correct the typos. We believe the manuscript and the figure captions have been substantially improved in the revision and is more readable.

9. The methods do not include sufficient details about how individual experiments were carried out.

Reply:

We have already added the suggested content to the manuscript on “Materials and Methods” in the revised manuscript. It mainly involves the following parts: ATPase activity assay (line 412-426), Microscale Thermophoresis (MST) assay (line 428-438), EM sample preparation and data collection (line 443-446), Liquid Chromatography with tandem mass spectrometry (LC-MS/MS) analysis (line 546-558), and added the section of cross-linking mass spectrometry (XL-MS) analysis (line 570-603).

Minor points:

1. A number of abbreviations are not spelled out when they first appear in the manuscript.

Reply:

We have carefully checked the manuscript and added the full descriptions of the abbreviations in the revised manuscript, for example hATP13A2 (line 44), SPG78, (line 46), and so on.

2. The term “electron density” is misused several times. The maps from cryo-EM reconstructions are not electron density maps.

Reply:

The misused terms have been corrected (e.g., line 108, 161, and 785).

3. Fig. 1f: should labels M1-10 be TM1-10?

Reply:

The labels have been corrected (Fig.1b in the revision).

4. Fig. 1h: are labels TM4 and TM5 swapped?

Reply:

We have corrected the error (Fig. 1h now has been moved to the supplementary Fig. 3h in the revised manuscript).

5. The term “stereo view” is used in the supplementary figure captions multiple times, but there are no stereo views.

Reply:

The term has been corrected in the revision, such as in Fig. 4d (line 688-689) and Supplementary Fig. 13 (now Supplementary Fig. 16) (line 868-869).

6. Line 136-137: “A blob of density corresponding to AMP-PCP (an ATP analog) was well solved near the KGxPE motif in the N domain (Fig. 1b)”. But the actual Fig. 1b does not show this.

Reply:

We have corrected this mistake. We have added Fig. R7 to illustrate the EM density of AMP-PCP.

Fig. R7 (Now Supplementary Fig. 3i) EM densities and atomic models of AMP-PCP.

7. Line 274 and 275: what are “-1e” and “-5e”?

Reply:

They represent the negative charges of phospholipids PS and PI which are -1 and -5, respectively. This has been clarified in the revision (line 232-233).

8. Line 650-651: “the atomic model ribbon is shown (grey), the protein depicted in ribbon representation (gray)”. It is unclear what this means.

Reply:

We have revised the figure legends (line 655-656).

Reviewer #2

- This manuscript by Mu et al. provides highly significant data and interesting interpretations for the field of the P-ATPases transporters particularly in the P5B subgroup which was shown to transport polyamines. This work advances the knowledge of the ATP13A2 which has been implicated in several neurological diseases.
- The manuscript focus on a hot area that is the structure of the protein ATP13A2. Other studies have been recently published on the same topic: Sim et al., Mol Cell, 2121;- Tillinghast et al., Mol Cell,2021; Chen et al., Cell Disc. 2021; Tomita et al., Mol. Cell 2021. These studies use essentially the same methodology that is the cryoEM structural analysis of the human ATP13A2, and together with the paper of Li et al. Nat. Commun. 2021 Jun 25, describing the structure of the yeast homolog yPK9, they provide a wealth of data on the structural features of the P5B-ATPases.
- Given the new data and the novel mechanisms proposed in my opinion this manuscript would require A) a more detailed description of the experimental conditions, B) a better explanation of the proposed interpretation of the data, C) a comparison with the results provided by recent studies. Each point is summarized below.

Mayor points

1. P16 line 469. Please indicate the composition of the ATPase reaction media, including the concentrations of Mg²⁺ and ATP. What is the enzyme specific activity in Fig1 Supp. For example in umol/mg.min. What is the meaning of about 50% ATPase of the WT when no SPM is added? How is this incorporated in the Albers-Post scheme?

Reply:

We thank the referee for their helpful suggestions. We have supplemented the composition of the ATPase reaction medium in the revised manuscript but the ATP and Mg²⁺ concentrations were not clear in the commercial kit. We have tried to contact the company but didn't get a clear answer. We have already updated Supplementary Fig. 1c, and the ATPase activity was displayed as μmol Pi liberated from ATP by 1 mg of ATP13A2 in 1 h (μmol Pi/mg/h).

We have corrected "WT" to "Control" in Supplementary Fig. 1c. We have no intention to describe the meaning of 50% ATPase of the WT when no SPM is added, and how is this incorporated in the Albers-Post scheme.

2. P16 line 497. Please describe the experimental media including ligands and protein concentration in the MST assays. How these media compare to the CryoEM?

Reply:

We have added more details of the MST assay conditions in the Material and Method section in the revised manuscript. (line 428-435). In the MST assay, the protein is labeled with a highly sensitive fluorophore at a concentration of only 10 nM; in contrast, the protein concentration is approximately 35-70 μ M in cryo-EM, which is much higher than in the MST assay. Similarly, the ligand concentration is much higher in cryo-EM (2 mM SPM) than the maximum concentration (50 μ M) in MST assay.

3. P17 line 507. Please provide a detailed composition of the media used for the each Cryo-EM structure, indicating concentration of protein, lipids, ions, nucleotides, nucleotide analogs, Pi analogs etc., incubation time and conditions before data acquisition.

Reply:

We have added the details of the cryo-EM sample preparation conditions for each intermediate state in the Material and method part in the revised manuscript. As follow: “...E1-ATP, 2 mM AMP-PCP; E1P-ADP, 5 mM NaF, 1 mM AlCl₃, 5 mM ADP; E1P, 10 mM NaF, and 2 mM AlCl₃; E2P, 10 mM NaF, and 2 mM BeSO₄; E2-Pi, 10 mM NaF, and 2 mM AlCl₃, 2 mM spermine; E2, 2 mM spermine. After incubation for 1 ~2 h on ice, the mixture was centrifuged at 13,000 rpm for 10 min at 4 °C, ...For E1-like state ATP13A2 (apo), freshly purified protein samples in SEC buffer were applied immediately.” (line 443-447). Concentration of protein and buffer as follow: “...SEC buffer containing 50 mM MOPS (pH 7.0), 100 mM KCl, 5 mM MgCl₂, 1 mM DTT, 0.06% digitonin ...concentrated to ~5-10 mg/mL before cryo-EM grid preparation.” (line 407-410).

4. P7 line 192. In addition to the SPM site 1 which was described in other studies, this manuscript describes novel extra binding pockets at site 2 and site 3 in the E2.Pi and E2 structures. This is a quite interesting observation. The E2 structure presented here, although at low resolution, is totally novel. However, the E2.Pi structure has been described in other studies and in contrast with the authors' findings, only SPM site1 was reported. In this regard, A) What action led the authors to see the other sites? B) Is there a P-ATPase in which the substrate has been observed in “transit” with several

binding sites in the same structure? Please explain and add to Discussion.

Reply:

We thank the referee for their positive comments and thoughtful considerations. Semi-quantitative analysis of SPM in the E2-Pi state (Fig. R3c) revealed that hATP13A2 in the E2-Pi state captured more SPM than the other intermediate states, including E1P-ADP, E2P, and E1-like (Fig. R3c). Structural analysis showed that the E1-like state presents a conformation similar to the E1-ATP state (Fig. R2a), in which Site₁ displayed a closed architecture without SPM. However, semi-quantitative analysis supported that the E1-like state contains high levels of SPM. All these incentivized us to believe the existence of extra binding sites for SPM, in addition to Site₁.

Next, we used cross-linking mass spectrometry to verify the potential binding sites of SPM (Fig. R4), and found that SPM could be covalently cross-linked with K506 (Fig. R5), an amino acid located at Site₂. This finding strongly supported that Site₂ is an SPM binding site. Mutations in Site₂ showed a decreased SPM-dependent ATPase activity. It was further confirmed that these sites might be related to the binding of SPM. Unfortunately, we have not got the direct evidence to support that the density in Site₃ was SPM derived; therefore, we have deleted Site₃ and the corresponding interpretation in the revised manuscript (line336-338).

However, substrates in “transit” with several binding sites in the same structure was not observed in other P-type ATPases. The following reasons may explain this: The P1, P2, and P3 ATPases are ion pumps whose substrates are small and very soluble, indicating the substrates would be released from the transmembrane regions once they enter the outer membrane leaflet. The P4 ATPases and P5A ATPases translocate lipids and transmembrane helix, respectively, which must be captured in a cavity with a large volume; however, the transmembrane regions' volume is too limitive to simultaneously bind multiple substrates with big size in different binding sites, which would distort the transport pathway. The ATP13A2 has a unique inward-facing cavity, composed of Site₂ and the release site, with a large volume, which provides an opportunity for binding an extra SPM (line 345-347).

5. Fig 3 shows a model of the protein in E2 structure with three SPM molecules. It can be interpreted as if the protein could bind three SPM molecules simultaneously. In contrast I noticed that the MST curves were fitted to single binding site model. A) What is the stoichiometry SPM/AT13A2 in E2? B) If E2P-E2 species have SPM bound in different sites there should be a difference in the binding energy. What is the fraction of protein with SPM bound at each site? Also, is difficult to see how this large substrate

would move through the protein without parallel changes in the protein.

Reply:

We appreciate the very inspiring questions, some of which have been partially answered in Q7 raised by referee#1 (Page 9).

Briefly, using the MST assay to determine the interaction stoichiometry requires high affinities between two molecules. Stoichiometry experiments are difficult for low-affinity interactions with K_d in the intermediate or high μM range since high concentrations are needed for both target and ligand. As a result, the Monolith detector is saturated due to high fluorescence. We used a mixture of labeled and unlabeled targets, but no desired result was obtained.

Therefore it is difficult to calculate the fraction of protein with SPM bound at each site. But we agree with the referee that the transmembrane domain would be rearranged to translocate the substrate across the membrane. However, cryo-EM structures solved in our work were snapshots of hATP13A2 in different intermediate states, making it difficult to see how this large substrate would move through the protein in successive conformational changes.

6. P10 line 303. The authors propose a mechanism of stimulation of ATP13A2 activity by PIP2 which is very different from that previously proposed for yPK9 by Li et al. Nat. Commun. 2021 Jun 25. A) Please discuss. B) How PIP2 affects SPM affinity? In the model proposed by the authors PIP2 accelerates K_{off} of SPM. Can the MTS experiment be done in the presence and in the absence of PIP2? Alternatively, how the dependence of ATPase activity with SPM is affected by PIP2?

Reply:

Besides being observed in our EM maps, the lipid band bound to the transmembrane domain was also observed in published works, three of which have proposed a model showing that lipid may promote the activity of hATP13A2²⁻⁴. However, the putative E2 structure only solved in our work showed an SPM-like density contacting lipid band, which illustrated how lipids may regulate the binding of SPM and thus the activity of hATP13A2; further, MD simulation analysis revealed that PIP2 could interact with the transmembrane domain with a relatively higher possibility (Fig. 4f, Supplementary Fig. 12 e and f). All these may provide a direct clue for explaining that lipid might regulate the activity of hATP13A2.

Tine et al. had already reported PIP2 stimulated the autophosphorylation of ATP13A2 at the biochemical level, resulting in the accumulation of EP state⁵. More interesting,

Veen et al. reported the addition of SPM stimulated dephosphorylation of hATP13A2, which promotes the accumulation of dephosphorylation state, whereas exogenous addition of PIP2 promoted SPM-dependent ATPase activity (Fig. 1e in Nature volume 578, pages 419-424 (2020))¹. Based on the above information, PIP2 and SPM can regulate phosphorylation and dephosphorylation, respectively, thereby accelerating the transport cycle of hATP13A2. Our biochemical assay revealed that ATP13A2 showed SPM-dependent ATPase activity (Fig. R6). Thus, combined with our structural information and MD simulation analysis, we proposed a model in which PIP2 regulates the activity of ATP13A2.

7. P12 line 371. The structural similarity in the E1P and E2P reported by the authors is at variance of previous studies of other P-ATPases. Furthermore, in the model presented in P26 line 691 the authors state “In the following transition of E1P to E2P, ADP release causes conformational rearrangements in the cytosolic domains and transmembrane regions, opening the entrance pocket ...”. Please explain.

Reply:

Based on the published data of flippase, adding phosphate analogs AlF_4^- to the reaction buffer could stabilize the P4-ATPase flippase in the E1P state^{6,7}. Thereby, to capture hATP13A2 in the E1P and E2P states, AlF_4^- and BeF_3^- were preincubated with hATP13A2, respectively, for one hour before vitrification. However, structural comparison showed the nominal E1P structure displayed a similar conformation to that of our E2P and published E2P structures (with a low RMSD of 0.81 Å, Fig. R2d). Thus, we are more certain that the nominal E1P structure that we resolved is indeed a structure of hATP13A2 in the E2P state. In the revised manuscript, we removed the information of the nominal E1P structure and the corresponding interpretation. Meanwhile, we removed the state “In the following transition of E1P to E2P, ADP release causes conformational rearrangements in the cytosolic domains and transmembrane regions, opening the entrance pocket ...”. (line 690-693 in the former version of manuscript)

8. P11 line 353. The concept of “buffering tank” is very appealing. However I am not aware of a similar proposal in other P-ATPase. Why ATP13A2 would be different in this respect? Could this be related to the SPM gradient between lysosome and cytosol and the direction of the transport?

Reply:

We performed the volume analysis of the cavities of intermediate states for P5A and P5B (Supplementary Fig. 12i and j). Results showed that the P5B subgroup has a relatively larger volume of the cavity; moreover, sequence alignment also showed the amino acids located at the cavity are highly conserved for P5B (Supplementary Fig. 9); all these indicate this cavity might be unique to P5B.

Previous studies have reported that ATP13A2 dysfunction leads to the lysosomal polyamine accumulation, and a reduced ATP13A2 activity lowers the cellular polyamine content and impairs cellular polyamine distribution⁹. Therefore, it is a good idea to speculate that the buffering tank might be related to the SPM gradient between lysosome and cytosol; however, we have no direct evidence to support this idea. The PPALP motif connecting TM4a and TM4b is a flexible loop that might act as a gating switch in-between the cytosol and the lysosome lumen, similar to the transport mechanism of Spf1⁸, preventing the polyamine back release, which defines the transport direction of ATP13A2.

Minor points

1. P16 line 471. “Follow the mixture”. Please rephrase

Reply:

We have re-written this sentence in the revised manuscript (line 423).

2. P5 lines 109-112. Perhaps is more adequate to indicate that this is true for P4 and P5 ATPases only and not for all P-ATPases.

Reply:

We have re-written this sentence in the revised manuscript. As follow: “Previous studies obtained high-resolution structures of other P-type ATPases in the E1 state by vitrifying the wild-type ATPase directly” (line 118-119).

3. P6 line 166. Perhaps the authors mean that “among P5B-ATPases members and not in P5A-ATPases.

Reply:

We have already rephased the sentence as follows: “The residues around Site₁, including W251, D254, D463, Q944, F963, and D967, are highly conserved among P5B ATPase members ” (line 156-157).

4. P7 line 179 “solvent explosion “? Do you mean exposure?

Reply:

We have already corrected the “solvent explosion” into “solvent exposure” in the revised manuscript (line 167).

5. Figure 1. Could you please add the transported substrate in the transport cycle?

Reply:

We have updated Fig. 1d according to the referee’s ideas.

6. Figure 2. Legend. “An illustrator”? Do you mean an illustration?

Reply:

We have removed the Fig. 2a after careful consideration.

Reviewer #3

The paper “Structural basis of a full conformational cycle of human ATP13A2 transporting polyamines” describes a polyamine transport pathway from lumen to cytosol based on multiple cryo-EM structures. Several structures are presented that confirm the overall architecture and polyamine entry on the luminal side and overall, the data appear solid. However, four papers (see below) from different labs have already reported multiple conformations of the same transporter studied here, ATP13A2, and as such the novelty of the data can be questioned. Further underscoring this, the authors refrain from doing comparisons to any of these previous studies. The authors have also not presented what is still missing in the field, following these previous achievements. This is important as it will highlight the potential significance of the current findings – what new information is the new data bringing to the field? Throughout the paper, it is not clear how the claimed conformations have

been stabilized and assigned in the transport cycle. Are the states inward or outward-facing, are the TM domains open or not, is substrate present or not and does the location change, which is intimately coupled to the E1-E1P-E2P-E2 cycle, or if they are similar to each other or to already available structures, which is relevant regarding the novelty. This is important also because many of the states reported in the recent papers are almost indistinguishable (even if different inhibitors have been supplemented), and hence they add little new information to regarding the biological or biochemical function of the protein. It is possible or perhaps even likely that such a scenario is also the case here, but I do not know as comparisons are missing. The authors own summary of their results in Fig. 5 suggests five different intermediates, but the abstract indicate that seven intermediates have been identified. The authors claim that they have uncovered a previously uncharacterized E2.

state of P5B-ATPase, with an inward-open cavity, also be present in other states, with two bound spermine molecules, despite the low overall resolution of that particular state renders modelling challenging. Also, inward-facing conformations are typically linked to E1 (inward-open) or E1P (inward-occluded) states in P-type ATPases and hence the authors likely need to revise how they assigned the new state. Combined with molecular dynamics simulations it is suggested that the surrounding phospholipids can interact with the polyamine to stimulate its release, which is an interesting model. It is also recommended that the paper is sent for language review before resubmission, as there are a large number of errors in the text. One such example is that it should be “transport pathway” not “transporting pathway”, which consistently incorrect throughout the text.

Specifications of the above-mentioned remarks:

1. The four previous multi-conformational structural studies on ATP13A2 (<https://pubmed.ncbi.nlm.nih.gov/34728622/>, <https://pubmed.ncbi.nlm.nih.gov/34715014/>, <https://pubmed.ncbi.nlm.nih.gov/34715013/>), <https://pubmed.ncbi.nlm.nih.gov/34798056/> and the related <https://pubmed.ncbi.nlm.nih.gov/34172751/> should be referenced and the findings summarized in the introduction. What are the missing gaps in the field?

Reply:

We thank the referee for their helpful suggestions. This question has been answered in the response to referee#1, accordingly, to describe the current knowledge of hATP13A2' structures, we have added some detailed descriptions on the previously published hATP13A2 structures and missing gaps to the introduction section of the revised manuscript (line 66-71).

2. It needs to be made clearer how do you mimic or stabilize the multiple states you have determined, and in the text, it must be made clearer which state the structure being referred to in any specific part of text is in, including what stabilizer is used. Details of which stabilizers were used, how and in which concentrations should be clarified in the methods. Currently there is one row (507) covering this, which is insufficient. Clarify and explain each configuration (e.g., inward open or outward open)? How do you distinguish these states? Different states mean different configurations (e.g., open/occluded) instead of relying on the different additives. Is the inward open cavity present in all states or a specific E2 state? If it is always open to the cytosol, how would polyamine back release be prevented?

Reply:

We thank the referee for their thoughtful considerations. We have added more details of the sample preparation conditions for each state in the Material and method section of the revised manuscript (line 443-447).

Human ATP13A2 transports SPM by coupling ATP hydrolysis with protein phosphorylation states and adopting a series of intermediate states, consisting of E1, E1-ATP, E1P-ADP, E1P, E2P, E2-Pi, and E2. Each intermediate state has their own configuration, with the SPM entering hATP13A2 in the E2P state from the lysosomal lumen, and exiting from hATP13A2 into the cytosol in the E2 state (Fig. R8 and

Supplementary Movie 1). Based on both the additives and their corresponding configurations, we assigned each structure along the transport cycle.

The inward-opening cavity was ubiquitous in all states and is formed mainly by TM2, TM4, and TM6. The PPALP motif connecting TM4a and TM4b is a flexible loop that might act as a gating switch in-between the cytosol and the lysosome lumen, similar to the transport mechanism of Spfl⁸, preventing the polyamine back release.

Fig. R8. (Now Fig. 1a) Schematic diagram of Post-Albers cycle for ATP13A2. The transport of SPM is shown in blue. Components that mimic the enzyme intermediates of their respective reaction cycles are shown in red. SPM is recruited from the lumen side of ATP13A2 in the E2P state and released into the cytoplasm in the E2 state.

3. The manuscript lacks comparisons, both with other available P5B structures, but also between the different structures in the current manuscript. How different are the different structures you have determined? How does the spermine compare between them? How does all this relate to previously determined structures? For example, it appears that you have two structures with spermine in site3, but there is no information on if the conformation of spermine is different in the two structures. What are the conformational changes between these states?

Reply:

We thank the referee for their helpful suggestions. This question has been partially answered in the response to referee#1 and we have added detailed structural comparisons between our structures and these published ones to the result and discussion section in the revision (Fig. R1 and R2) (line126, 130, 140, 320, 324, 327, and 329).

In addition, we have turned down the statement of the Site₃ and redefined it as a releasing site in the revised manuscript.

For convenience, the structural comparisons for each intermediate state are summarized as follows:

- E1-like State:

We captured the E1-like cryo-EM structure of WT ATP13A2 without adding ligands. In contrast, Sim et al.'s obtained the nominal E1-apo state (Protein Data Bank (PDB): 7N76 and 7N75) by introducing two mutations (D458N/D962N) to stabilize the ATP13A2 in E1-like state. Works from Tomita et al.² and Tillinghast et al.⁴ showed no E1 state structures. Moreover, since the E1 state is dynamic due to no additives, it is a little weird that Chen et al.⁹ obtained the high-resolution structure of the nominal E1 state (PDB: 7FJM) by processing 3, 000 micrographs of wild-type hATP13A2.

The structural comparison showed that the E1-like structure in our work is different from the reported E1 structures (PDB: 7N76, 7N75 and 7FJM; Fig. R2b).

- E1-ATP State:

For E1-ATP States, the overall structure is highly similar to deposited PDB: 7VPI (root mean square deviation (RMSD)=1.11 Å), PDB:7N74 (RMSD=1.41 Å) and PDB:7M5V (RMSD=1.07 Å) in published papers (Fig. R1).

- E1P-ADP State:

For E1P-ADP States, the overall structure is highly similar to that of PDB:7VPJ (RMSD=1.26 Å), PDB:7N73 (RMSD=1.45 Å) and PDB:7FJP (RMSD=0.98 Å) in published papers (Fig. R1).

- The nominal E1P State:

To capture the E1P and E2P states, phosphate analogs AlF_4^- and BeF_3^- were preincubated with hATP13A2 on ice, respectively, for one hour before vitrification. Interestingly, we found both structures are very similar (with a low RMSD of 0.81 Å; Fig. R2c). In addition, structural comparison among published E2P structures (PDB: 7VPK, 7N70 and 7M5X) made us believe that the nominal E1P captured in our work is an E2P structure. Therefore, all information deduced from the nominal E1P structure were modified in the revised manuscript.

The reported E1P-like structure (PDB: 7N77) was obtained through introducing two-point mutations in hATP13A2, which present an obviously different conformation from the structure of the nominal E1P obtained by our experiment (RMSD=6.22 Å; Fig. R2e). It is speculated that the conformational difference may be caused by point mutations.

- E2P State:

All the reported E2P states were captured by adding BeF_3^- , and the E2P structures (PDB: 7VPK and 7M5X) displayed the similar conformation to our E2P structure.

- E2-Pi State:

The E2-Pi structure in our work displayed a highly similar overall conformation to all published E2-Pi structures of ATP13A2.

- Putative E2 State:

Published articles have not yet reported any E2 cryo-EM structures of ATP13A2. After

processing more than 10 thousand EM micrographs, we obtained a putative E2 structure, presenting a widely open cavity towards the cytosol, which displayed a quite different conformation from all reported hATP13A2 structures.

Minor points:

Introduction

1. Row 52: It should be “Kufor-Rakeb”, not “Kufor-Rekab”

Reply:

We have already corrected the “Kufor-Rekab” into “Kufor-Rakeb” in the revised manuscript (line 45).

2. Row 53: What is SPG78? Why have you explained what Kufor-Rakeb is, but not this one?

Reply:

We have added the clarification of SPG78 in the revised manuscript (line 46).

3. Row 60: How is Mn²⁺ homeostasis related to ATP13A2 and polyamine transport?

Reply:

We have no intention to discuss how the Mn²⁺ homeostasis is related to ATP13A2 and polyamine transport. We have removed the sentence that “Overexpression of human ATP13A2 homolog in *C. elegans* could reduce Mn²⁺ caused cellular toxicity” (line 59-61 in the former version of manuscript) to address your concerns and hope that the manuscript is more readable.

4. Row 65: Include more details about the different states. How are these intermediate states stabilized? What is different between each state, which ones are opened toward lumen or cytosol and which are closed, for example? It would be a benefit for people who are not familiar with P-type ATPases.

Reply:

P-type ATPases can be trapped at various stages of the cycle by combining ion binding with ATP analogs (such as AMP-PCP or AMP-PNP) and structural analogs of phosphorylation, phosphoryl transfer, or phosphate release (BeF₃⁻, AlF₄⁻, MgF_x)

respectively¹⁰. Here, we report the cryo-EM structures of the hATP13A2 in its six distinct intermediates: an apo state (E1-like), the nonhydrolyzable ATP analog β , γ -methyleneadenosine 5'-triphosphate (AMP-PCP)-bound state (E1-ATP), the adenosine diphosphate-inorganic phosphate (ADP-Pi) analog AlF_4^- -ADP-bound state (E1P-ADP), the BeF_3^- -bound phosphoenzyme ground state (E2P), and the AlF_4^- -bound dephosphorylation state with the substrate SPM (E2-Pi), and the substrate SPM-bound state (putative E2 state), throughout the Post-Albers cycle.

We have added a diagram to illustrate the transport process of polyamines, favoring our understanding which states are outward facing the lumen and which are inward facing the cytosol (Fig. R8).

5. Row 68: This sentence makes it sound like P5s are unique among P-types because they are only found in eukaryotes and they have subfamilies, which is incorrect.

Reply:

We have rephrased the sentence to make it clearer in the revised manuscript. As follows: “The P1, P2, and P3 ATPases are ion pumps; the P4 subfamily comprises lipid flippases; and the P5 ATPases, which are found only in eukaryotic species, are further divided into type A and B subgroups” (lines 54-57).

6. Row 73 (and elsewhere): It appears that P1-P5 are referred to as subfamilies, but P5A-P5B are also referred to as subfamilies. Please use different terms for them to avoid confusion.

Reply:

We have used the “subgroup”¹¹ to rephrase the sentence and make it clear in the revised manuscript (line 57).

7. Row 75: What was presented in these papers needs to be expanded upon. Also, what does it mean that they “were online”? They have been published.

Reply:

We have rephrased the sentence to describe these papers and added more details about the reported structures in suitable part of the revised manuscript (line 66-68).

8. Rows 79-86: This paragraph should be expanded. Describe the N-terminal region, as it has not been introduced before? What is known about the interaction with lipids?

How do they interact?

Reply:

As the referee suggested that it is important to describe the N-terminal region. It has been reported that ATP13A2 accumulates in an inactive autophosphorylated state and that N-terminal hydrophobic domain specifically recognizes signaling lipids PA and PI (3, 5) P2 which are able to stimulate the autophosphorylation of ATP13A2^{1,5}, and the interaction between N-terminal region and lipid was extensively studied in previous studies⁵.

9. Rows 98-99: Is it possible to understand the pathogenic mechanism only based on the structure and functional analysis of ATP13A2? How is drug development relevant to the current paper, as small molecules rarely can increase function?

Reply:

Based on the structural and biochemical analysis of ATP13A2, we can propose a mechanism underlying ATP13A2 transporting polyamines from lysosomes to the cytoplasm and maintaining the homeostasis of polyamine, which helps us understand how disease-related mutations on the ATP13A2 protein impact the transport efficiency of polyamines.

We also highlighted disease-causing mutations of ATP13A2 within the atomic model in Supplementary Fig. 16i, disease-causing mutations mostly located at the center of functional activity, and amino acids that interact with SPM. Structurally, it helps us understand the pathogenic mechanism of ATP13A2.

Structure-based drug discovery is an important technique for designing and optimizing innovative drugs. Mutations of ATP13A2 is highly related to early-onset Parkinson's disease and SPG78, and the high-resolution structures could provide a clue to designing drugs targeting ATP13A2.

Results

10. Row 105-106: Please comment on your high levels of ATP hydrolysis in the absence of added cargo.

Reply:

It is widely accepted that substrates could stimulate the ATP hydrolysis activity of P-type ATPases^{10,12}; mass spectrometric analysis of our purified hATP13A2 protein showed endogenous SPM were co-purified with hATP13A2 (Supplementary Fig. 1e),

the endogenous SPM would increase the levels of ATP hydrolysis. This may explain why we had high levels of ATP hydrolysis in the absence of added cargo.

11. Overall structure section: considering the overall structure of hATP13A2 has been described several times before, this is not a novel finding and could be considerably condensed. You may consider moving parts of it to the introduction.

Reply:

We agreed with the referee's opinion. Following the referee's suggestions, we have moved parts of the content in the overall structure section to the introduction. We have also detailed the progress and gaps in studying the structures of hATP13A2 in the introduction section of the revised manuscript (line 66-73).

12. Row 156 (and row 258, among others): It occasionally sounds like ATP13A2 has a channel structure instead of a transporter. For example, what do you mean by a "channel-shaped architecture" in row 156? Or that "several channels were observed to connect to site3 in row 258? Because it is not a channel.

Reply:

To avoid the misunderstanding, we have revised the manuscript and used the chalk-shaped architecture to describe the binding pocket for SPM (line 147-148) and the binding pocket (line 159).

13. Row 156: What does "where substrates dominate" refer to? That there is substrate bound in the entry site? Or that the concentrations of polyamines is higher in the lumen than cytosol? But why would it then need a transporter?

Reply:

We're grateful for the reviewers' helpful questions. What we want to interpret is that Site₁ captured the SPM. Accordingly, we rephrased the relevant content in the revised manuscript. As follow: "....., had a chalk-shaped architecture open to the luminal side where substrates were captured."(line147-158). We have no intention to describing the concentrations of polyamines in the lumen and cytosol.

14. Row 165 and figure S2: ATP13A1 is a P5A ATPase, not P5B, which explains the poor sequence homology!

Reply:

We thank the referee for pointing out this mistake. We have removed the relevant content in the revised manuscript.

15. Row 190-192: What does this sentence mean? I feel particularly confused about “would subject SPM to encompass the hATP13A2 extensively”.

Reply:

To make it clear, we have rewritten this sentence as follows: “By adding exogenous SPM into the E2-Pi cryo-EM samples, we captured substrate binding at multiple putative binding sites ...” (line 176-178).

16. Polyamine movement section: What is function of site2? Does it exist in other P5B-type ATPases?

Reply:

To confirm the function of Site₂, we introduced two-point mutations in Site₂, and the subsequent SPM-dependent ATPase activity assay showed these point mutations greatly decreased the activity of hATP13A2, which highly suggested Site₂ might be an intermediate binding site for SPM and plays an essential role in the transport of SPM. We are also interested in whether Site₂ exists in other P5B-type ATPases which may also involve in the transport of SPM. Amino acid sequence alignment showed amino acids at Site₂ are highly conserved among P5B-type ATPases, which leads us to speculate that Site₂ might exist in other P5B-type ATPases (Supplementary Fig. 9).

17. Row 247: How does this compare to the other structure with spermine in site3?

Reply:

We have not got the direct evidence to support the density in Site₃; therefore, we have deleted Site₃ and the corresponding interpretation in the revised manuscript. Moreover, we have added the following discussion in the revised manuscript (line 336-338).

18. Row 255-277: Why are these negatively charged phospholipids detected by MD close to site3, but not site2?

Reply:

We have not got the direct evidence to support the density is SPM-derived in Site₃. To

make it more accurate, we removed the description of the SPM-binding property of Site₃ in the revised manuscript. The separation movement of TM2 and TM4b made a large space for releasing polyamines. Surface charge analysis showed positively charged amino acids are highly distributed on the surface near the release site (Supplementary Fig. 16h). We speculated that these positively charged amino acids may attract the negatively charged phospholipids near the release site, which was partially supported by the results of the MD simulation (Supplementary Fig. 12e-g). Meanwhile, the negatively charged phospholipids near the release site may contribute to the release of polyamines into the cytoplasm (Fig. 4d-f and Supplementary Fig. 12h). Till now, we have no definite interpretation why phospholipids were not detected by MD close to Site₂. However, we speculate that Site₂ is inaccessible by lipids due to the chemical environment.

19. Row 262: This makes it sound like the water-filled cavity is new for this structure, but have you not already described it?

Reply:

The interpretation of this sentence was deduced from the information of Site₃ described in the former manuscript. Unfortunately, we have no direct evidence to support that Site₃ was an SPM binding site; therefore, we turn down the interpretation of Site₃ and remove the sentence “This makes it sound like the water-filled cavity is new for this structure”.

Discussion

20. Rows 367-372: This belongs in results.

Reply:

We have removed our manuscript in the revised manuscript.

Methods

21. Row 434: Here it is stated that a catalytically dead mutant is generated, but it is not referenced anywhere in the text. There is also no information in the methods of how the other mutants were generated, and specifically which mutants they were.

Reply:

We have rephrased the sentence in the revised manuscript. The details are as follows: “Its point mutants were generated with the Fast Mutagenesis System kit (Transgen,

FM111), and sequences were verified by Sanger DNA sequencing.” (line 3781-382).
"a catalytically dead mutant" is explained in Supplementary Fig. 1g legend , as follows:
D513N, the catalytically dead mutant hATP13A2, served as a positive control (line 724).

22. Row 440: Just before transfection, the cells were transfected?

Reply:

To address the reviewer’s concern, we have rewritten this sentence as follows: “... just before cell transfection, cells were seeded to...” (line 386-387).

23. Row 491: Just at the start of the row “1% to% (0-1.5 min). What is the second % ?

Reply:

We have updated the sentence in the revised manuscript. As follow: “: ... 1% to 1% (0-1.5 min), ...” (line 568).

24. Row 525: It is stated that more than 3500 movies were collected for each structure, but based on Table 1 that is not true for E1P-ADP and E1-ATP.

Reply:

We have rephrased the sentence as follows: “The number of micrographs collected for each stated grid is described in Supplementary Table 1.”(line 466-467).

25. Row 530: Figures S5 to S11 are reference. Should it not be figures S3 to S9?

Reply:

We have corrected it.

Figures

26. The following figures are never referenced, and should be removed or be referenced to: 1h, 2a and 2c.

Reply:

We have removed 2a and 2c, and moved Fig. 1h to the supplementary Fig. 3h in the revised manuscript.

27. The supplementary figures do not appear in the order that they are first referenced in the text. Based on the text, the supplementary figures should have the following order: S1, S3-9, S2, S11-12, S10 and S13.

Reply:

We have rearranged supplementary figures in a new order. The supplementary figures S3-9, S2, S11, and S12 are rearranged as S2-8, S9, S12, and S15, respectively. In addition, we have removed the S10.

28. The following figures need their color schemes clarified/adjusted: 1a should state what the color scheme is in the legend: which colors are used for which parts? In fig 3a the spermines themselves are colored, not be pockets. Also, spermine in site3 is not magenta. S2 states that cyan is the least conserved, but all cyan regions are completely conserved. In S10a the side chains appear to have different colors than the helices they are part of, which does not seem to be the case in S10c. How is the surface colored in S10e and f? In S11d and S12b there are no empty histograms, which the legend states.

Reply:

We sincerely thank the referee for kindly reminding and helpful suggestions. As suggested by the reviewer, we have added more details of color scheme in Fig.1 legend (line 635-640). We have corrected the “black represents the most conservative, followed by red, then the weakest is cyan” into “cyan represents the most conservative, followed by black, then the weakest is red” in Supplementary Fig. 2 (now Supplementary Fig. 9) states (line 789-790). The empty histograms in the Supplementary Fig. 11d and Supplementary Fig. 12b (now Supplementary Fig. 12g and Supplementary Fig. 15f) legends have been corrected to solid-colored histograms (line 825 and 863). To make the story clear, we have removed the Fig. 3a and the Supplementary Fig. 10 in the former version of manuscript. Moreover, since we have not got the direct evidence to support the density is SPM-derived in Site₃, we removed the description of the SPM-binding property of Site₃ in the revised manuscript.

29. Fig 1h: Why is this part of a main figure?

Reply:

Fig. 1h has been moved to the supplementary Fig. 3h in the revised manuscript.

30. Fig 2a: Why is this necessary?

Reply:

In the revised manuscript, we have removed Fig. 2a.

31. Figs 2j and S1: The text is too small to be readable in all (2j) or parts of (S1) these figures.

Reply:

We have adjusted the fonts for better visibility.

32. Fig 6: Why does this figure have 5 models, when you claim to present 7 different states? Presenting E2 in two rotated forms is unnecessary in this kind of scheme.

Reply:

As mentioned in the revised manuscript, the E1P-ADP structure has a similar conformation to the E1-ATP state (RMSD=0.72 Å), and the nominal E1P structure is in the E2P state. Therefore, we used five structures to propose a model of ATP13A2 transporting polyamine. Also, we removed the E2 rotated forms in Fig. 6 in the revised manuscript.

33. Fig S4h: There is no figure S4h.

Reply:

We updated the manuscript to make it more readable in the revised manuscript.

34. Fig S10h: Why is this part of figure S10? Would it not be more appropriate in figure S1?

Reply:

We have re-numbered Supplementary Fig. 10h to Supplementary Fig. 1f in the revised manuscript.

35. Fig S12a: Which state(s) did you use to generate these?

Reply:

In the revised manuscript, we have rearranged Supplementary Fig. 12a as

Supplementary Fig. 15a in the new order, and a figure legend is described in more detail as follow: Supplementary Fig. 15 Lipid binding to hATP13A2 in the E2-Pi and putative E2 state obtained from CG MD simulations. (line 856-857).

Reviewer #4

Jianqiang Mu et al present a structural study of the human ATP13A2 polyamine transporter. By using cryo-EM they solve the structure of this transmembrane protein in seven different conformational states. The cryo-EM structures are accompanied by molecular dynamics simulations and functional studies. All together, the study allows to rationalize the functional cycle of this protein, including the steps in which it undergoes major conformational rearrangements to enable the access and egress of polyamine molecules, and remarkably the putative involvement of lipids.

I find this is an extremely interesting study which substantially advances our understanding of the polyamine transport across lysosome membranes. The possibility to capture the protein under different conditions provides a complete structural picture of this important process.

I have the following comments

1. It is unclear to me how the (un)binding of ATP, ADP and Pi at the cytosolic units drives the seen conformational rearrangements at the transmembrane side. The manuscript largely focuses on the structural rearrangements at the transmembrane segments, which is of course relevant as there is where the polyamine molecules enter and exit. However the paper does not mention in sufficient detail how these changes connect to the ATP (and ADP+Pi)-dependent conformational changes of the A,N, and P cytosolic units. As the study nicely solved the structure of the whole protein, and in different states, it will be extremely interesting to check the connection between what is happening at the catalytic sites and the accessibility of the transmembrane sites.

Reply:

We thank the referee for their insightful suggestions. We totally agree that it will be extremely interesting to see how the (un)binding of ATP, ADP, and Pi at the cytosolic domains drives the conformational rearrangements at the transmembrane domain. The long-range communication between the cytosolic domains and the TM domains ensures the directional pumping and an economical use of ATP. Based on the most well-studied member of the P-type ATPase family, sarco-/endoplasmic reticulum Ca^{2+} -ATPase (SERCA), it has been proposed that the energy stored in the stretched TM3 linker might serve as a spring which can be used as a driving force for the rotation of the cytosolic domains, such as in the E1P \rightarrow E2P transition. However, it is still very challenging to

provide dynamic molecular details to connect all the intermediate states to explain the coupling between ligand binding and the long-range communication between the cytosolic domains and the TM domains. In principle, this could be addressed by MD simulations. But so far, even for the extensively studied P2-type ion transporter, there are rare reports. We will have to leave this question for further studies. But in the revision, we have added more descriptions and discussions on the conformational rearrangements of both the cytosolic domains and the TM domain between different intermediate states and also the differences from the recently published structures. We hope this may partially answer the questions.

2. In particular, I wonder to what extent the here-determined structures explain how the release of ADP opens the (distant) site 1 to enable polyamine entrance and how the release of Pi connects sites 1 with 2 and 3 to facilitate polyamine passage.

Reply:

We thank the referee for their fantastic question. All these structures described in this manuscript were determined by the single-particle cryo-EM technique, which could solve the structure of proteins with a relatively stable and static conformation. Therefore, only based on the static structural information, it is not straightforward to answer to what extent the here-determined structures explain how the release of ADP opens the (distant) Site₁ to enable polyamine entrance and how the release of Pi connects Site₁ with Site₂ and release site to facilitate polyamine passage. But we expect these structures open the gate to explain the mechanism in more details in complement with MD simulations with the help of enhanced sampling techniques.

3. The authors used MD simulations to check if the S2 and S3 sites could be potentially filled by lipid molecules. They did not observe that. The observation that negatively-charged phospholipids specifically bind to the protein to promote substrate release could have been elaborated further with this technique. The egress of polyamine molecules from the sites via the cytosolic lipid leaflet, relevant during the transport cycle, could have also been monitored in the simulations to support the lipid-mediated polyamine-release hypothesis.

Reply:

We agree that the phospholipids binding and substrate release could be studied using the MD simulations. We indeed have been performing further coarse-grained simulations to investigate the release of polyamine, but still without sufficient statics

due to the slow timescale of the process. We are currently collecting more and longer trajectories to get robust statistics. Moreover, we are also performing all-atom MD simulations to provide further support and atomic details to the polyamine-release process. This will take substantial computational time and resource to finish. Therefore, we plan to elaborate the substrate release mechanism in a separate manuscript.

Besides the simulations of the putative E2 state, we further supplemented the simulations on the E2-Pi state without SPM to check if both sites can be occupied by lipids. Similar to what we observed in the putative E2 state, the results revealed that no phospholipids were observed in the E2-Pi state as shown in the simulated density map of lipids, therefore providing further support that the density captured in the cryo-EM map is more likely to be from SPM (Fig. R9).

Figure R9 (now Supplementary Fig. 14 c, d): A left sectional view (c) and a top view (d) of the average lipid density maps obtained from coarse-grained MD simulations in E2-Pi state without SPM, respectively.

4. The Authors indicate that structures were available for other members of this ATPase family and even for this protein in few of the states. A comparison with these previous structures will help to realize conserved-structural features across the family.

Reply:

We thank the referee for this great suggestion which has also been raised by referee#1. We have done careful comparisons with these recently published structures (Fig. R1 and R2) and discussed the conserved-structural features in the revision (line126, 130, 140, 320, 324, 327, and 329).

5. lines 261-263: The mentioned water-filled cavity can not be seen in the Figure S11h. Maybe a zoom to the region of interest will help here.

Reply:

A zoom-in view of the water-filled cavity has been added (Fig.R10).

Fig. R10 (Now Supplementary Fig. 12c) The averaged density map of water obtained from the all-atom MD simulation of ATP13A2 in the putative E2 state is shown as magenta meshes.)

Minor:

1. Fig.s 2a and 5a not mentioned in the text. The scheme in Fig 6 is very useful and could be used throughout all the figures to highlight the states that are being presented in the context of the whole transporting cycle.

Reply:

We have removed Fig. 2a and 5a in the revised manuscript. We agree with the referee that Fig. 6 is useful. To make the manuscript more readable and avoid redundancy, we added Fig 1a and modified Fig 1d to highlight the states along the transporting cycle.

2. Sub figures are not ordered according to their mentions in the text (example: figures 2de-g are presented first than fig 2b-c). Reading flow will benefit of figures being presented in ordering number.

Reply:

We have reordered supplementary figures. The supplementary figures S3-9, S2, S11, and S12 have been rearranged as S2-8, S9, S12, and S15, respectively. And S10 was removed in the revised manuscript.

3. Abstract and line 51: "hATP13A2". "h" of human? has not been defined before.

Reply:

Yes, "h" stands for human. We have clarified it in the revision (line 44).

4. Line 65: what do E1 and E2 stand for?

Reply:

We used similar E1 and E2 definitions as Sarco-/endoplasmic reticulum Ca^{2+} -ATPase (SERCA) which is the most extensively studied member of the P-type family¹³. Here E1 represents the major state of P-type ATPases with the substrate binding site(s) open to the cytoplasm and E2 is the other major state with the substrate binding site(s) facing the cytoplasmic side.

5. Fig.s 1f-h may work better as Fig.s 1a-c to help readers to understand the overall protein topology, before entering into the details of the structures and the cycle.

Reply:

We have updated Fig. 1 in the revised manuscript.

Reference:

- 1 Van Veen, S. *et al.* ATP13A2 deficiency disrupts lysosomal polyamine export. *Nature* **578**, 419-424, doi:10.1038/s41586-020-1968-7 (2020).
- 2 Tomita, A. *et al.* Cryo-EM reveals mechanistic insights into lipid-facilitated polyamine export by human ATP13A2. *Molecular cell* **81**, 4799-4809.e4795, doi:10.1016/j.molcel.2021.11.001 (2021).
- 3 Sim, S. I., von Bülow, S., Hummer, G. & Park, E. Structural basis of polyamine transport by human ATP13A2 (PARK9). *Molecular cell* **81**, 4635-4649.e4638, doi:10.1016/j.molcel.2021.08.017 (2021).
- 4 Tillinghast, J., Drury, S., Bowser, D., Benn, A. & Lee, K. P. K. Structural mechanisms for gating and ion selectivity of the human polyamine transporter ATP13A2. *Molecular cell* **81**, 4650-4662.e4654, doi:10.1016/j.molcel.2021.10.002 (2021).
- 5 Holemans, T. *et al.* A lipid switch unlocks Parkinson's disease-associated ATP13A2. *Proceedings of the National Academy of Sciences of the United States of America* **112**, 9040-9045, doi:10.1073/pnas.1508220112 (2015).
- 6 Hiraizumi, M., Yamashita, K., Nishizawa, T. & Nureki, O. Cryo-EM structures capture the transport cycle of the P4-ATPase flippase. *Science (New York, N.Y.)* **365**, 1149-1155, doi:10.1126/science.aay3353 (2019).
- 7 Nakanishi, H. *et al.* Transport Cycle of Plasma Membrane Flippase ATP11C by Cryo-EM. *Cell reports* **32**, 108208, doi:10.1016/j.celrep.2020.108208 (2020).
- 8 McKenna, M. J. *et al.* The endoplasmic reticulum P5A-ATPase is a transmembrane helix dislocase. *Science (New York, N.Y.)* **369**, doi:10.1126/science.abc5809 (2020).
- 9 Chen, X. *et al.* Cryo-EM structures and transport mechanism of human P5B type ATPase ATP13A2. *Cell discovery* **7**, 106, doi:10.1038/s41421-021-00334-6 (2021).
- 10 Dyla, M., Kjaergaard, M., Poulsen, H. & Nissen, P. Structure and Mechanism of P-Type ATPase Ion Pumps. *Annual Review of Biochemistry* **89**, 583-603, doi:10.1146/annurev-biochem-010611-112801 (2020).
- 11 De La Hera, D. P., Corradi, G. R., Adamo, H. P. & De Tezanos Pinto, F. Parkinson's disease-associated human P5B-ATPase ATP13A2 increases spermidine uptake. *The Biochemical journal* **450**, 47-53, doi:10.1042/bj20120739 (2013).
- 12 Bai, L. *et al.* Structural basis of the P4B ATPase lipid flippase activity. *Nature communications* **12**, 5963, doi:10.1038/s41467-021-26273-0 (2021).
- 13 Palmgren, M. G. & Nissen, P. P-type ATPases. *Annual review of biophysics* **40**, 243-266, doi:10.1146/annurev.biophys.093008.131331 (2011).

REVIEWER COMMENTS

Reviewer #1 (Remarks to the Author):

The authors have addressed the reviewer's comments by including new data and/or revising the manuscript.

Reviewer #2 (Remarks to the Author):

In this revised version of the manuscript by Mu et al., the authors have added new experimental data and more detail in the methodology, and in consequence, the revised version has improved considerably compared with the first version of the work. However, in my opinion, some of the original doubts about the proposed interpretations are still not solved and the new data generate more questions. Fundamentally more attention should be given to the presentation of the data. I would suggest a good language and style correction of the whole manuscript. With the aim of possibly helping the authors, some points are indicated below although the list is not exhaustive.

1- Supplementary Figure 1c. Does the new manuscript present data from new measurements or they are the same data as in the original manuscript? When new data is added the authors should explicitly indicate so. Comparing the Methods section of the original and the new versions of the manuscript is clear that ATPase activity was measured by using a different kit and under different conditions. Was the original data replaced by new measurements? It seems awkward that neither with the first kit nor with the second the authors are able to provide a detailed composition of the reaction media. It is necessary to know the concentration of ATP in the reaction media and whether the liberation of Pi was linear with time in order to interpret the activity data.

Also according to Methods, it seems that all the ATPase measurements were done in the absence of any added lipids. Most P-ATPases are inactive or rapidly inactivated by detergents in this condition. The lack of lipids during the ATPase assays is not consistent with the important role that the authors are proposing for PIP2.

2-. I would appreciate it if in the text of the manuscript the authors could moderate the subjective judgment of their own results. I found that the word "dramatic" is used 7 times in the text plus

“dramatically” 2 times. In my opinion, this is unjustified. For example line 97 it is stated that “the purified hATP13A2 protein exhibited high ATP hydrolysis activity, which was dramatically increased by the presence of polyamines”. The value of ATPase activity that the authors are reporting is in the same range as some other previous studies (van Veen et al., 2020) and the increase at saturating SPM is about 1.8 fold (Figure 1G), which is lower than in other publications.

3- As I have already pointed out, about 50 % of the hATP13A2 ATPase measured by the authors is polyamine independent (Figure 3E and Supplementary Figure 1C). Although in their rebuttal the authors indicate that they “have no intention to describe the meaning of 50% ATPase of the WT when no SPM is added”, it seems to me that this issue deserves at least a comment in the text, first because is at variance with previous studies of the same protein but also because it is related to the interpretation of the new structural models provided by the authors. Indeed, Reviewer#3 (criticism n10) raised a similar concern “Please comment on your high levels of ATP hydrolysis in the absence of added cargo.” The authors’ response to Reviewer#3 indicates that the polyamine-independent activity is due to the presence of co-purifying polyamines. Assuming that that is the case, as suggested by the detection of SPM by MS, is not obvious to me why SPM would resist the purification protocol and why the endogenous SPM would allow continuous ATPase cycling without diffusing out of the detergent micelle.

In Supplementary Figure 1G, the results are expressed as % of WT basal. What is the meaning of WT basal? I guess that by “basal” the authors mean without SPM? However, the legend of the figure indicates that the measurements were done in the presence of 100 uM SPM, so it makes no sense to express them as % of WT without SPM. Figure 3E shows that the ATPase activity of mutants F241A and F419A is near that of the WT in the absence of SPM, and is not stimulated by SPM. However, in Figure 1G the same mutants have about 40 % of the “WT basal”. Since they are not stimulated by SPM, how could the basal activity (without added SPM) be due to endogenous SPM?

4- The meaning of the results presented in Supplementary Fig 13 is not clear. Why the “E1-like” form would contain SPM? What is the control? Is the detected SPM specific? Is the higher content of SPM of the E2P just because it is the only condition containing 2 mM SPM before SEC?

5-According to the legend of Supplementary Fig 9 showing the multiple alignments of P5 ATPase sequences, Site2 residues are marked by a red asterisk. D515 which is the site of formation of the catalytic EP has a red asterisk. Does this mean that SPM interacts directly with D515 at Site 2?

Reviewer #3 (Remarks to the Author):

While I think that the revised version of the manuscript has been much improved technically, my main critique against the manuscript remains. Although the breadth and quality of the data is impressive, I am not convinced that they significantly advance the field. From the revised version and the responses to the reviewers' questions, it is now clear that the majority of the structurally determined intermediates have already been deduced (even of the same target) by other groups. Specifically, in the response to the reviewers, the authors state that the obtained E1-ATP, E1-ADP, 'nominal' E1P, E2P and E2-Pi states, all are highly homologous to already available structures (the 'nominal' E1P is apparently an E2P state). This matter, how similar the determined structures are to already available structural information, was raised by three separate reviewers.

The two remaining structures, the E1-like and the putative E2 states, were determined at low resolution (5.7 and 4.9 Å resolution, respectively), rendering it difficult to assign non-main chain features of the cryo-EM density. In this perspective, the assignments of SPM and PIP₂ are not unambiguous in the putative E2 state (Fig. 4). I am also not convinced by the novelty of the E1-like state. Based on the RMSD comparisons shown in Fig. S10, the E1-like state appears to be as similar/different to available structures as the other states determined in the manuscript that, according to the authors, are highly similar to already determined structures. Indeed, Fig. S11 suggests the E1-like and the E1-ATP state are highly similar. In contrast, there is no doubt that the putative E2 state has not been reported previously! Furthermore, I would emphasize polyamine binding Site₂, as it may be the first-time substrate is detected in the release pathway (although one would have expected to see this in later stage of the transport cycle).

I am also questioning how much the field will learn on the function of the P5(B)-ATPases with the data presented in the manuscript. It appears to me that the outline of the polyamine transport pathway (including the inward-facing pocket supported by polyamine binding Site₂) as well as the lipid-dependence proposed by the authors (Fig. 6) have already been described in the five papers that were indicated by several of the reviewers (see for example the transport models proposed in Fig. 7 of PMID 34715013 and in Fig. 7 of PMID 34715014). This is perhaps also reflected in the revised abstract of the manuscript, where no new information on the function of hATP13A2 is indicated, other than "... suggesting a potential polyamine transport pathway", something that has already been reported on in the already available five papers on structures of P5B-ATPases.

Nonetheless, the manuscript has been substantially improved. My previous minor concerns have been resolved, maybe apart from the aspect that I am not convinced that the generated structures will be useful for drug-design. The diseases associated with ATP13A2 are loss-of-function disorders, and hence it cannot be expected that structure-based design of small molecules will alleviate the symptoms of the affected patients.

Minor comment:

Page 7, rows 236-246: Is this in agreement with how the cytosolic domains move in other P-type ATPases?

Reviewer #4 (Remarks to the Author):

The authors have adequately addressed my concerns. The manuscript has substantially improved after this revision.

Reviewer #5 (Remarks to the Author):

The authors present an interesting dissection of ATP13A2's mechanism of polyamine transport. I am not familiar with the field, but I have two general comments before my assessment of the mass spectrometry data:

1. The success of this approach is anchored to the validity of the different biochemical treatments to “freeze out” the various intermediates. This seems to be based on prior work. There are some functional validations (i.e. the increased ATPase activity with the addition of spermine) but some of the conditions to transition from one state to the next look very subtle (for example the difference between E1P and E2P). How do we know that these formulations induce conformations consistent with transport? It seems to me that the phosphorylation status could at least be confirmed (e.g. autophosphorylation and dephosphorylation). For the non-expert, it would be useful to indicate why these biochemical manipulations are valid for such studies.
2. At times it is unclear what is meant by inward vs outward. It seems the authors use inward for cytosolic facing (e.g. “inward opening cavity on the cytosolic side”), when I suspect they mean lumen-side? This should be clarified in the text.

Regarding the MS-based crosslinking analysis, the data are not very convincing:

1. A complete description of the results is lacking. I doubt very much that only one spectrum resulting from labeling was obtained. In these experiments, the reagent is incubated for a very long time with the protein and spermine, very likely leading to extensive labeling of both. It looks like only one result – the favorable one – was selected. I'm sure this is not the case, but all the data need to be presented.

2. Even if this is the only detectable spectrum, it does not mean spermine is located at Site2. I question the approach in general. It is entirely likely that spermine is labeled on one amine independently of the protein, and then goes on to label the protein with the remaining activated ester. The labeling reaction can be non-specific, in other words. I suspect it is, given that the crosslinking reagent is much larger than spermine, and the binding site is crowded. There is a cluster of lysines around the putative Site2, which seems odd given that electrostatics of binding would be highly unfavorable.

3. The authors could improve the analysis by:

a. Repeating the crosslinking experiment with the functionally dead form (i.e. the Y240A and F419A mutants) and determine if crosslinking is affected.

b. Present the MS data (with scores and cutoffs) for all the reaction products in both cases, and not just K506.

c. Clarify exactly how the database searches were done. The spectral annotations in Supplementary Figure 14 look very poor. I doubt these are correct. An entrapment analysis would be useful here, to verify.

d. Provide annotated spectra to an accepted proteomics repository.

4. There are other matters that need to be addressed regarding the MS-related content of the manuscript:

a. "Solved" should be "dissolved" (Lines 555ff)

b. Was the quantitation of spermine collected on a Quantiva or an Exactive? Both are listed (lines 561ff)

c. The description of an MS curve desorption line temperature is foreign to me. Do you mean the temperature of the desolvation gas?

d. There are two redundant sections of text: lines 577-580 and lines 588-590.

e. On line 602, why are three exclusion times listed?

f. The fragmentation "pathways" presented in Supplementary Figure 1 are highly simplified and unclear. The source of m/z 112 is a further fragmentation of m/z 129. Either present the full fragmentation pathway, label the MS2 spectrum with the fragment structures, or reference a "fit" to some digital repository of metabolite MS2 data.

Response to referees' comments

We thank the referees again for their valuable time and constructive suggestions. After careful consideration of their comments, we have prepared a clearer, more thorough revision that we hope will address the remaining concerns. In particular, we have repeated the cross-linking experiment, which showed good consistency and confirmed the presence of the extra substrate binding sites in the cytosolic cavity. Please find below a point-by-point response to the referees with our responses in **Blue** and the referees' comments in **Red**.

Reviewer #1 (Remarks to the Author):

The authors have addressed the reviewer's comments by including new data and/or revising the manuscript.

Reply:

We thank the reviewer for his/her previous feedback and assistance in making the manuscript better.

Reviewer #2 (Remarks to the Author):

In this revised version of the manuscript by Mu et al., the authors have added new experimental data and more detail in the methodology, and in consequence, the revised version has improved considerably compared with the first version of the work. However, in my opinion, some of the original doubts about the proposed interpretations are still not solved and the new data generate more questions. Fundamentally more attention should be given to the presentation of the data. I would suggest a good language and style correction of the whole manuscript. With the aim of possibly helping the authors, some points are indicated below although the list is not exhaustive.

Reply:

We would like to thank the reviewer for their evaluation and constructive feedback.

1- Supplementary Figure 1c. Does the new manuscript present data from new measurements or they are the same data as in the original manuscript? When new data is added the authors should explicitly indicate so. Comparing the Methods section of the original and the new versions of the manuscript is clear that ATPase activity was measured by using a different kit and under different conditions. Was the original data replaced by new measurements? It seems awkward that neither with the first kit nor with the second the authors are able to provide a detailed composition of the reaction media. It is necessary to know the concentration of ATP in the reaction media and whether the liberation of Pi was linear with time in order to interpret the activity data.

Also according to Methods, it seems that all the ATPase measurements were done in the absence of any added lipids. Most P-ATPases are inactive or rapidly inactivated by detergents in this condition. The lack of lipids during the ATPase assays is not consistent with the important role that the authors are proposing for PIP2.

Reply:

We are grateful that the reviewer brought this to our attention. In the revised manuscript, we have clarified the kit and conditions that were used. The data in Supplementary Fig. 1g are the same data that were presented in the original manuscript. They were generated by measuring ATPase activity with an imported kit containing 1 mM ATP (<https://www.bioassaysys.com/datasheet/DATG.pdf>). The data in Fig. 3e (line 204, 205) and Supplementary Fig. 1c (line 102), obtained using a domestic kit containing 2 mM ATP (line 414), showed that the liberation of Pi was linear over time, consistent with

the results obtained with the imported kit. The updated data and results associated with Fig. 3e and Supplementary Fig. 1c are highlighted in green in the revision (line 100-102, line 204-206, and line 411-425).

We agree with the reviewer that detergent can deactivate ATPases, but our results showed that the purified ATP13A2 was indeed active in detergent. This is supported by recently reported results for ATP13A2 from other research groups obtained using detergent¹⁻³. Although we did not add lipids prior to measurement, we detected a clear lipid density in the cryo-EM density maps, suggesting the presence of endogenous lipids that may be important to the structure and function of ATP13A2. The lipid density was further proposed to be PIP2 based on the results of our coarse-grained MD simulations. Indeed, previous studies have demonstrated that adding PIP2 may promote SPM-dependent ATPase activity⁴.

2-. I would appreciate it if in the text of the manuscript the authors could moderate the subjective judgment of their own results. I found that the word “dramatic” is used 7 times in the text plus “dramatically” 2 times. In my opinion, this is unjustified. For example line 97 it is stated that “the purified hATP13A2 protein exhibited high ATP hydrolysis activity, which was dramatically increased by the presence of polyamines”.

The value of ATPase activity that the authors are reporting is in the same range as some other previous studies (van Veen et al., 2020) and the increase at saturating SPM is about 1.8 fold (Figure 1G), which is lower than in other publications.

Reply:

We thank the referee for their comments and have extensively revised the manuscript to provide more objective descriptions.

We are grateful to the reviewer for pointing out the difference between the ATPase activity values measured in our study and those reported in previous publications. The difference may result from the use of different experimental conditions. Veen et al. measured the activity of ATP13A2 embedded in microsomes, whereas we used a detergent-based assay. We observed a 2.4-fold (rather than 1.8-fold) increase in ATPase activity in the presence of saturating SPM, similar to the increase reported by Tomita et al.⁵. Therefore, while the experimental conditions may account for the difference in the activity measurements, we observed the same overall trend and our conclusions are unaffected.

3- As I have already pointed out, about 50 % of the hATP13A2 ATPase measured by the authors is polyamine independent (Figure 3E and Supplementary Figure 1C). Although in their rebuttal the authors indicate that they “have no intention to describe the meaning of 50% ATPase of the WT when no SPM is added”, it seems to me that this issue deserves at least a comment in the text, first because it is at variance with previous studies of the same protein but also because it is related to the interpretation of the new structural models provided by the authors. Indeed, Reviewer#3 (criticism n10) raised a similar concern “Please comment on your high levels of ATP hydrolysis in the absence of added cargo.” The authors’ response to Reviewer#3 indicates that the polyamine-independent activity is due to the presence of co-purifying polyamines. Assuming that that is the case, as suggested by the detection of SPM by MS, is not obvious to me why SPM would resist the purification protocol and why the endogenous SPM would allow continuous ATPase cycling without diffusing out of the detergent micelle.

In Supplementary Figure 1G, the results are expressed as % of WT. What is the meaning of WT basal? I guess that by “basal” the authors mean without SPM? However, the legend of the figure indicates that the measurements were done in the presence of 100 μ M SPM, so it makes no sense to express them as % of WT without SPM. Figure 3E shows that the ATPase activity of mutants F240A and F419A is near that of the WT in the absence of SPM, and is not stimulated by SPM. However, in Figure 1G the same mutants have about 40 % of the “WT basal”. Since they are not stimulated by SPM, how could the basal activity (without added SPM) be due to endogenous SPM?

Reply:

To address this important point, we have provided additional evidence that endogenous SPM can resist the purification protocol, and that the co-purified SPM increased the levels of ATP hydrolysis. We also performed LC-MS/MS experiments to semi-quantitatively analyze the capture of SPM by hATP13A2 in different intermediate states. The results showed that hATP13A2 in all intermediate states, including E1P-ADP, E2P, and E1-like (Supplementary Fig. 13c), contains high levels of endogenous SPM. Moreover, our E2P structure features a density corresponding to SPM in Site₁, which has also been observed in the reported structures of hATP13A2 proteins purified using the same detergent⁶. Collectively, these findings support the ability of SPM substrates to resist the purification protocol and allow continuous ATPase cycling. In addition, the cryo-EM maps of hATP13A2 displayed a density band corresponding to lipids in the TM region. The combination of the endogenous SPM and TM region-associated lipids could increase the ATPase activity of wild-type hATP13A2.

The data labeled "% of WT basal" represent the ratio of the ATPase activity of the mutant ATP13A2 over that of wild-type ATP13A2 in the presence of 100 μ M SPM. We apologize for the confusing labeling and have removed the term "basal" from Supplementary Fig. 1g in the revised version.

Fig. 3e shows that the ATPase activities of mutants F240A and F419A are comparable to that of the wild-type form in the absence of SPM. However, the ATPase activities of mutants F240A and F419A are not enhanced by adding SPM. This finding is consistent with the results shown in Supplementary Fig. 1g, that the same mutants displayed approximately 40% of the activity of wild-type ATP13A2 in the presence of 100 μ M SPM.

Single point mutations do not disrupt the overall volume and shape of Site₂, so the substrate-binding capacity of Site₂ is retained and the initial ATPase activities of the mutants F240A and F419A are comparable to that of wild-type ATP13A2. However, the mutation of Site₂ residues may affect the translocation of SPM, resulting in a failure of the mutants to exceed basal ATPase activity levels in the presence of exogenous SPM.

4- The meaning of the results presented in Supplementary Fig 13 is not clear. Why the "E1-like" form would contain SPM? What is the control? Is the detected SPM specific? Is the higher content of SPM of the E2P just because it is the only condition containing 2 mM SPM before SEC?

Reply:

To exclude contamination of the sample preparation buffer with non-specific SPM, we used size exclusion chromatography (SEC) buffer as a control to prepare MS samples. No SPM signals were detected by LC-MS/MS analysis of samples prepared in SEC buffer. As described in Supplementary Fig. 13a and 13b, the E2P, E1-like, and E1P-ADP states were prepared without the addition of exogenous SPM. However, both LC-MS/MS analysis and cryo-EM consistently suggested that hATP13A2 in the E2P state contains endogenous SPM. Although the cryo-EM map of hATP13A2 in the E1-like form was obtained at relatively low resolution, precluding assignment of an exact density to SPM, LC-MS/MS analysis revealed that hATP13A2 in the E1-like form contained high levels of endogenous SPM. Therefore, we think the SPM observed in the E1-like state is specific.

Based on the questions raised by the reviewer, we speculated that their concerns related to the high content of SPM in the E2-Pi state (rather than the E2P state). To address this concern, we included a control sample (100 μ L SEC buffer containing 2 mM SPM) and

performed SEC under equivalent conditions. The counterpart fraction was also subjected to semi-quantitative analysis by LC-MS/MS. We found one order of magnitude less SPM in the counterpart fraction than in the E2-Pi state. Given that the free SPM has been removed by SEC in the E2-Pi state, we believe that the higher SPM content resulted from exogenous SPM that specifically bound to ATP13A2 in the E2-Pi state.

5. According to the legend of Supplementary Fig 9 showing the multiple alignments of P5 ATPase sequences, Site2 residues are marked by a red asterisk. D515 which is the site of formation of the catalytic EP has a red asterisk. Does this mean that SPM interacts directly with D515 at Site 2?

Reply:

We thank the referee for pointing out this mistake. D513 (rather than D515) located on the P-domain, but was mislabeled as a Site₂ residue. The red asterisk for D513 has been removed from Supplementary Fig. 9.

Reviewer #3 (Remarks to the Author):

While I think that the revised version of the manuscript has been much improved technically, my main critique against the manuscript remains. Although the breadth and quality of the data is impressive, I am not convinced that they significantly advance the field. From the revised version and the responses to the reviewers' questions, it is now clear that the majority of the structurally determined intermediates have already been deduced (even of the same target) by other groups. Specifically, in the response to the reviewers, the authors state that the obtained E1-ATP, E1-ADP, 'nominal' E1P, E2P and E2-Pi states, all are highly homologous to already available structures (the 'nominal' E1P is apparently an E2P state). This matter, how similar the determined structures are to already available structural information, was raised by three separate reviewers.

The two remaining structures, the E1-like and the putative E2 states, were determined at low resolution (5.7 and 4.9 Å resolution, respectively), rendering it difficult to assign non-main chain features of the cryo-EM density. In this perspective, the assignments of SPM and PIP2 are not unambiguous in the putative E2 state (Fig. 4). I am also not convinced by the novelty of the E1-like state. Based on the RMSD comparisons shown in Fig. S10, the E1-like state appears to be as similar/different to available structures as the other states determined in the manuscript that, according to the authors, are highly similar to already determined structures. Indeed, Fig. S11 suggests the E1-like and the E1-ATP state are highly similar. In contrast, there is no doubt that the putative E2 state has not been reported previously! Furthermore, I would emphasize polyamine binding Site2, as it may be the first-time substrate is detected in the release pathway (although one would have expected to see this in later stage of the transport cycle).

I am also questioning how much the field will learn on the function of the P5(B)-ATPases with the data presented in the manuscript. It appears to me that the outline of the polyamine transport pathway (including the inward-facing pocket supported by polyamine binding Site2) as well as the lipid-dependence proposed by the authors (Fig. 6) have already been described in the five papers that were indicated by several of the reviewers (see for example the transport models proposed in Fig. 7 of PMID 34715013 and in Fig. 7 of PMID 34715014). This is perhaps also reflected in the revised abstract of the manuscript, where no new information on the function of hATP13A2 is indicated, other than "... suggesting a potential polyamine transport pathway", something that has already been reported on in the already available five papers on structures of P5B-ATPases.

Nonetheless, the manuscript has been substantially improved. My previous minor concerns have been resolved, maybe apart from the aspect that I am not convinced that the generated structures will be useful for drug-design. The diseases associated with ATP13A2 are loss-of-function disorders, and hence it cannot be expected that structure-based design of small molecules will alleviate the symptoms of the affected patients.

Reply:

We appreciate the reviewer's kind comment about the breadth and quality of our data, and their recognition of the novelty and importance of the putative E2 state we solved. We also agree with the reviewer that most hATP13A2 disease-related mutations result in a loss of function. However, we believe the generated structures provide a clearer understanding of the molecular mechanism of hATP13A2 transport, which will be necessary to guide rational drug design, even if we do not yet anticipate how the mechanistic findings will be used.

Regarding the novelty of our work, as the reviewer stated, the structures we solved cover nearly the full conformational cycle of a P5 ATPase, including the first E2 structure. The novel E2 structure allows us to reveal previously unappreciated conformational rearrangements in the cytosolic and TM domains, which are essential to understanding the substrate-releasing mechanism of hATP13A2 and the transport mechanisms of other P-type ATPases.

Moreover, we report polyamine binding at multiple sites, which has not been reported in previous studies. We captured multiple substrate binding sites distributed along the TM regions, suggesting a potential substrate transport pathway. Of note, we identified an always-open inward-facing cytosolic cavity that may serve as a substrate buffering tank. This feature distinguishes hATP13A2 from other P-type ATPase subfamilies and suggests it utilizes a unique substrate-transporting mechanism. Furthermore, we integrated cryo-EM densities with MD simulations to reveal a featured belt of negatively charged phospholipids around the substrate exit site, which may play an active role in promoting substrate release. In summary, while our findings are consistent with those of previous studies, they advance our knowledge of hATP13A2 recruitment, transport, and release of polyamine substrates.

Minor comment:

Page 7, rows 236-246: Is this in agreement with how the cytosolic domains move in other P-type ATPases?

Reply:

The cytosolic domains of ATP13A2 and other P-type ATPases members—such as ATP8A1⁷ in P4-ATPase lipid flippases, Spf1⁸ in P5A-ATPase, and SERCA⁹ in P2-ATPase cation pumps—rearrange in a similar manner during the Post-Albers cycle.

Reviewer #4 (Remarks to the Author):

The authors have adequately addressed my concerns. The manuscript has substantially improved after this revision.

Reply:

We thank the reviewer for his/her previous feedback and assistance in making the manuscript better.

Reviewer #5 (Remarks to the Author):

The authors present an interesting dissection of ATP13A2's mechanism of polyamine transport. I am not familiar with the field, but I have two general comments before my assessment of the mass spectrometry data:

1. The success of this approach is anchored to the validity of the different biochemical treatments to “freeze out” the various intermediates. This seems to be based on prior work. There are some functional validations (i.e. the increased ATPase activity with the addition of spermine) but some of the conditions to transition from one state to the next look very subtle (for example the difference between E1P and E2P). How do we know that these formulations induce conformations consistent with transport? It seems to me that the phosphorylation status could at least be confirmed (e.g. autophosphorylation and dephosphorylation). For the non-expert, it would be useful to indicate why these biochemical manipulations are valid for such studies.

Reply:

We thank the referee for their comments. The P-type ATPases translocate various substrates by coupling ATP hydrolysis with protein phosphorylation and dephosphorylation, and adopting a series of intermediate states, consisting of E1, E1-ATP, E1P-ADP, E1P, E2P, E2-Pi, and E2, in a process known as the Post-Albers reaction¹⁰. It is well accepted that P-type ATPases can be trapped at various stages of the cycle with ATP analogs (such as AMP-PCP or AMP-PNP) and structural analogs of phosphorylation, phosphoryl transfer, or phosphate release such as BeF_3^- , AlF_4^- , or MgF_x , respectively¹¹. Here, we report the cryo-EM structures of hATP13A2 in its six distinct intermediates, including the phosphorylated E2P state. We also tried to capture the phosphorylated E1P state using phosphate analog AlF_4^- ; however, the final conformation of the nominal E1P state is quite similar to that of the E2P state (RMSD=0.82 Å). To date, one E1P-like structure of hATP13A2 has been reported by introducing two points mutations⁶. Thus, it is very challenging to capture the true E1P state of hATP13A2.

2. At times it is unclear what is meant by inward vs outward. It seems the authors use inward for cytosolic facing (e.g. “inward opening cavity on the cytosolic side”), when I suspect they mean lumen-side? This should be clarified in the text.

Reply:

We thank the referee for their thoughtful comment. We used Fig. 1a and 1d to illustrate the polyamine transport process, allowing us to highlight which states face outward toward the cavity and which face inward toward the cytosol.

Regarding the MS-based crosslinking analysis, the data are not very convincing:

1. A complete description of the results is lacking. I doubt very much that only one spectrum resulting from labeling was obtained. In these experiments, the reagent is incubated for a very long time with the protein and spermine, very likely leading to extensive labeling of both. It looks like only one result – the favorable one – was selected. I'm sure this is not the case, but all the data need to be presented.

Reply:

We thank the referee for their helpful suggestions. To address the reviewer's concern, we have repeated the cross-linking experiments with their comments in mind and all of the results are presented in the revised manuscript. In terms of the experimental design, we added a SEC purification step to remove free SPM before preparing the cross-linked samples.

2. Even if this is the only detectable spectrum, it does not mean spermine is located at Site2. I question the approach in general. It is entirely likely that spermine is labeled on one amine independently of the protein, and then goes on to label the protein with the remaining activated ester. The labeling reaction can be non-specific, in other words. I suspect it is, given that the crosslinking reagent is much larger than spermine, and the binding site is crowded. There is a cluster of lysines around the putative Site2, which seems odd given that electrostatics of binding would be highly unfavorable.

Reply:

We understand the reviewer's concern, which we have addressed by performing the cross-linking experiments again. The complete cross-linking data are reported in Supplementary Data 1. To reduce the non-specific cross-linking of SPM to ATP13A2, an additional SEC step was performed to remove free SPM before preparing the cross-linked samples. In the repeated cross-linked MS experiments, better-quality MS/MS spectra were obtained and several sites—including K420, K506, and K432—were found to be cross-linked to SPM. These findings indicate that the cross-linking experiment has good repeatability, and that the cross-linking reactions of SPM to K506, K420, and K432 were specific.

In addition, we examined the cryo-EM structure of the E2-Pi state. The structural analysis showed that the amino acids contacting the SPM substrate within 5 Å are mainly hydrophobic residues, including Y240, F428, V429, M477, P470, and L499 (Fig. 3c). The distances from K506, K420, and K432 to SPM were 5.3 Å, 6.3 Å, and 9.4 Å, respectively (Fig. 3d). Taken together, these findings indicate that the SPM in Site₂ is captured mainly through hydrophobic interactions rather than unfavorable electrostatic forces. This conclusion is further supported by the MD simulation results, which show that SPM can be stably captured in Site₂.

Given the configuration and dynamic properties of BS³, its arm length ranges from 2.0 Å to 11.4 Å, spanning the distances between K506, K420, and K432 and the SPM in Site₂. In addition, the amine groups of K506, K420, K432, and SPM are all located in the peripheral region of the membrane protein ATP13A2, allowing BS³ to form links between K506, K420, K432, and SPM.

We supplemented the complete cross-linking data in Supplementary data 1, which showed that the residues cross-linked with the SPM (BS3+SPM (340.2838)) were mainly K506, K240, and K243. Even if there were occasional amino acids mono-linked or loop-linked to BS3, such as K150 and K404, these residues were distributed on the surfaces of cytosolic domains, which are far away from the substrate transport pathway. These amino acids were labeled by BS3 possibly due to the highly active property of BS3, having no relationship of SPM.

3. The authors could improve the analysis by:

- a. Repeating the crosslinking experiment with the functionally dead form (i.e. the Y240A and F419A mutants) and determine if crosslinking is affected.
- b. Present the MS data (with scores and cutoffs) for all the reaction products in both cases, and not just K506.
- c. Clarify exactly how the database searches were done. The spectral annotations in Supplementary Figure 14 look very poor. I doubt these are correct. An entrapment analysis would be useful here, to verify.
- d. Provide annotated spectra to an accepted proteomics repository.

Reply:

We thank the reviewer for these constructive suggestions.

- a. Based on the reviewer's comments, we repeated the cross-linking experiments for wild-type hATP13A2 and found that, in addition to K506, K420 and K432 were also cross-linked to SPM, suggesting good repeatability and confirming that SPM was specifically captured in Site₂. The ATPase assay showed that the Y240A variant had

relatively low ATPase activity; however, the binding pocket is unlikely to be disrupted by a single point mutation. Therefore, we believe that the Y240A mutant may have the same cross-linking spectrum as the wild-type ATP13A2.

b. We have compiled the complete cross-linking mass spectra data for wild-type ATP13A2 in Supplementary Data 1, according to your suggestion.

c. We have updated the detailed database search method in the Methods section (line 597-609) and updated Supplementary Fig. 14d.

d. We have submitted the raw MS files (submission size: 1.1 G) together with a description of their content and the reporting template to the PRIDE database (accession number: PXD037493) (<https://www.ebi.ac.uk/pride/archive/>).

Supplementary Fig. 14 d Annotated MS2 spectrum of the protein hATP13A2 (Q9NQ11) peptide INLGGKLQLVCFDK.

4. There are other matters that need to be addressed regarding the MS-related content of the manuscript:

a. “Solved” should be “dissolved” (Lines 555ff)

b. Was the quantitation of spermine collected on a Quantiva or an Exactive? Both are listed (lines 561ff)

c. The description of an MS curve desorption line temperature is foreign to me. Do you mean the temperature of the desolvation gas?

d. There are two redundant sections of text: lines 577-580 and lines 588-590.

e. On line 602, why are three exclusion times listed?

f. The fragmentation “pathways” presented in Supplementary Figure 1 are highly simplified and unclear. The source of m/z 112 is a further fragmentation of m/z 129. Either present the full fragmentation pathway, label the MS2 spectrum with the fragment structures, or reference a “fit” to some digital repository of metabolite MS2 data.

Reply:

- a.** We have corrected this mistake in the manuscript. (line 554)
- b.** To make the protocol clear, we have rewritten this sentence as follows: “LC-MS/MS analysis was carried out on a Thermo LC-MS Q Exactive-Orbitrap system with a Kinetex 1.7 μm C18 100 Å LC column (100 \times 2.1 mm, Phenomenex).” (line 560-561).
- c.** We apologize for the lack of clarity. We have rewritten this sentence as follows: “MS curve desorption line (CDL) was set at 250 °C...”. (line 564-565)
- d.** We have removed redundant sentences from the revised manuscript. (line 579)
- e.** We have corrected the exclusion times in the revised version. (line 595)
- f.** We thank the referee for their helpful suggestions and have updated Supplementary Fig. 1e accordingly.

Reference

- 1 Tillinghast, J., Drury, S., Bowser, D., Benn, A. & Lee, K. P. K. Structural mechanisms for gating and ion selectivity of the human polyamine transporter ATP13A2. *Mol. Cell* **81**, 4650-4662.e4654, doi:10.1016/j.molcel.2021.10.002 (2021).
- 2 Li, P., Wang, K., Salustros, N., Grønberg, C. & Gourdon, P. Structure and transport mechanism of P5B-ATPases. *Nat. Commun.* **12**, 3973, doi:10.1038/s41467-021-24148-y (2021).
- 3 Chen, X. *et al.* Cryo-EM structures and transport mechanism of human P5B type ATPase ATP13A2. *Cell Discov.* **7**, 106, doi:10.1038/s41421-021-00334-6 (2021).
- 4 van Veen, S. *et al.* ATP13A2 deficiency disrupts lysosomal polyamine export. *Nature* **578**, 419-424, doi:10.1038/s41586-020-1968-7 (2020).
- 5 Tomita, A. *et al.* Cryo-EM reveals mechanistic insights into lipid-facilitated polyamine export by human ATP13A2. *Mol. Cell* **81**, 4799-4809.e4795, doi:10.1016/j.molcel.2021.11.001 (2021).
- 6 Sim, S. I., von Bülow, S., Hummer, G. & Park, E. Structural basis of polyamine transport by human ATP13A2 (PARK9). *Mol. Cell* **81**, 4635-4649.e4638, doi:10.1016/j.molcel.2021.08.017 (2021).
- 7 Hiraizumi, M., Yamashita, K., Nishizawa, T. & Nureki, O. Cryo-EM structures capture the transport cycle of the P4-ATPase flippase. *Science* **365**, 1149-1155, doi:10.1126/science.aay3353 (2019).
- 8 McKenna, M. J. *et al.* The endoplasmic reticulum P5A-ATPase is a transmembrane helix dislocase. *Science* **369**, doi:10.1126/science.abc5809 (2020).
- 9 Sørensen, T. L., Møller, J. V. & Nissen, P. Phosphoryl transfer and calcium ion occlusion in the calcium pump. *Science* **304**, 1672-1675, doi:10.1126/science.1099366 (2004).
- 10 Kühlbrandt, W. Biology, structure and mechanism of P-type ATPases. *Nat. Rev. Mol. Cell Biol.* **5**, 282-295, doi:10.1038/nrm1354 (2004).
- 11 Dyla, M., Kjaergaard, M., Poulsen, H. & Nissen, P. Structure and Mechanism of P-Type ATPase Ion Pumps. *Annu. Rev. Biochem.* **89**, 583-603, doi:10.1146/annurev-biochem-010611-112801 (2020).

REVIEWER COMMENTS

Reviewer #2 (Remarks to the Author):

The authors have corrected the mistakes and have added arguments that favor their conclusions. I believe the manuscript contains interesting data and interpretations and can be accepted for publication.

Reviewer #5 (Remarks to the Author):

The authors have provided extensive updates. I have restricted my comments to reviewer #5 matters.

1. It is still unclear what is meant by "inward vs. outward", as their new figure 1 additions confuse me more. The lumen IS the cytosolic side.
2. The addition of SEC to remove excess SPM is a good idea, and seemed to improve the quality of the one reported crosslink. The K_d looks to be about 1 micromolar, so a bit weak for such an approach, but possible. I can accept the results. However, they report seeing linkages to two other lysines from this redo, but I only see one linkage in the supplementary material they provide. This should be fixed. It is unfortunate they chose not to test the catalytically less active mutants, but I accept their point.
3. I still don't know what an MS curve desorption line is. They will need to clarify. I think they mean the desolvation region, involving the transfer capillary, and that is at 250C?
4. There are still other solved/dissolved errors.

Response to referees' comments

We thank the referees again for their valuable time and constructive suggestions. After careful consideration of their comments, we have prepared a clearer, more thorough revision that we hope will address the remaining concerns. Please find below a point-by-point response to the referees with our responses in **Blue** and the referees' comments in **Red**.

Reviewer #2 (Remarks to the Author):

The authors have corrected the mistakes and have added arguments that favor their conclusions. I believe the manuscript contains interesting data and interpretations and can be accepted for publication.

Reply:

We would like to thank the referee for all the comments on our manuscript, which helped us to refine the story more smoothly.

Reviewer #5 (Remarks to the Author):

The authors have provided extensive updates. I have restricted my comments to reviewer #5 matters.

1. It is still unclear what is meant by "inward vs. outward", as their new figure 1 additions confuse me more. The lumen IS the cytosolic side.

Reply:

Sorry for the confusion. "Inward" means the cytosolic side, and "outward" means the luminal side, as shown in Fig. 1a. We have clarified this in the manuscript.

2. The addition of SEC to remove excess SPM is a good idea, and seemed to improve the quality of the one reported crosslink. The Kd looks to be about 1 micromolar, so a bit weak for such an approach, but possible. I can accept the results. However, they report seeing linkages to two other lysines from this redo, but I only see one linkage in the supplementary material they provide. This should be fixed. It is unfortunate they chose not to test the catalytically less active mutants, but I accept their point.

Reply:

We thank the referee for their valuable suggestions. We have added Supplementary Fig. 14 e to show that K420 was cross-linked to the SPM. We did not provide the spectra of the K432 cross-linked SPM because of the poor quality of the spectra, and raw data are available in supplementary data 1.

Supplementary Fig. 14e Annotated MS2 spectrum of the protein hATP13A2 (Q9NQ11) peptide PINFKFYKHSMK.

3. I still don't know what an MS curve desorption line is. They will need to clarify. I think they mean the desolvation region, involving the transfer capillary, and that is at 250C?

Reply:

We have rephrased the sentence to make it clearer in the revised manuscript. As follows: “The capillary temperature was set at 320 °C and aux gas heater temperature was set at 370 °C”. (line563-565).

4. There are still other solved/dissolved errors.

Reply:

We thank the referee for pointing out this mistake. Based on your comments, we have re-written this sentence as follows: “hATP13A2 proteins were extracted with acetonitrile ... ” (line 554). “... sample was re-dissolved with...” (line 557).